# Remote sensing of ocean surface currents: A review of what is being observed and what is being assimilated

Jordi Isern-Fontanet[1,2], Joaquim Ballabrera-Poy[1], Antonio Turiel[1,2], and Emilio García-Ladona[1]

[1]Institut de Ciències del Mar (CSIC), Passeig Marítim de la Barceloneta 37-49, E-08003 Barcelona, Spain
[2]Barcelona Expert Center in Remote Sensing (CSIC), Passeig Marítim de la Barceloneta 37-49, E-08003 Barcelona, Spain
*Correspondence to:* Jordi Isern-Fontanet (jisern@icm.csic.es)

**Abstract.** Ocean currents play a key role in Earth's climate, they impact almost any process taking place in the ocean, and are of major importance for navigation and human activities at sea. Nevertheless, their observation and forecasting are still difficult. First, no observing system is able to provide direct measurements of global ocean currents at synoptic scales. Consequently, it has been necessary to use Sea Surface Height and Sea Surface Temperature measurements and refer to dynamical frameworks to derive the velocity field. Second, the assimilation of the velocity field into numerical models of ocean circulation is difficult mainly due to lack of data. Recent experiments assimilating coastal-based radar data have shown that ocean currents will contribute to increase the forecast skill of surface currents, but require to be applied in multi-data assimilation approaches to better identify the thermohaline structure of the ocean. In this paper we review the current knowledge on these fields and provide global and systematic view on the technologies to retrieve ocean velocities in the upper ocean and the available approaches to assimilate this information into ocean models.

## 1 Introduction

Surface ocean currents contribute to characterize the Earth's climate (WMO, 2015). Knowledge of ocean surface velocities is a key and cross-cutting issue that impacts on many societal challenges far beyond the research context in geophysical fluid dynamics. As such, ocean surface currents have been included in the list of essential climate variables (Bojinski et al., 2014). Indeed, ocean currents transport and redistribute heat, dissolved salts, sediments, plankton, nutrients and ocean pollutants. Strong ocean currents define corridors used by marine mammals, birds and fishes, and sustain their migration in search for food, breeding sites and spawning areas. As a result, knowledge of the detailed structure and variability of ocean currents is required for fisheries and environmental management. Furthermore, surface currents directly affect many important socio-economic activities as global maritime trade and shipping or marine pollution and safety, to mention a few.

Ocean surface currents are the result of a non-trivial combination of different types of periodic and aperiodic phenomena whose ranges span a continuous spectra of space and time scales, from basin-wide motions ($\sim$1000 km) to fast narrow currents and mesoscale eddies (30-100 km wide), submesoscale features (1-10 km) and quasi-three-dimensional turbulence scales (1-100 m). Due to the complexity of the currents power spectra, the meaning and representativeness of any velocity average (and the corresponding residual current) is a function of the averaging period and region, and its time and location (Neumann, 1968).

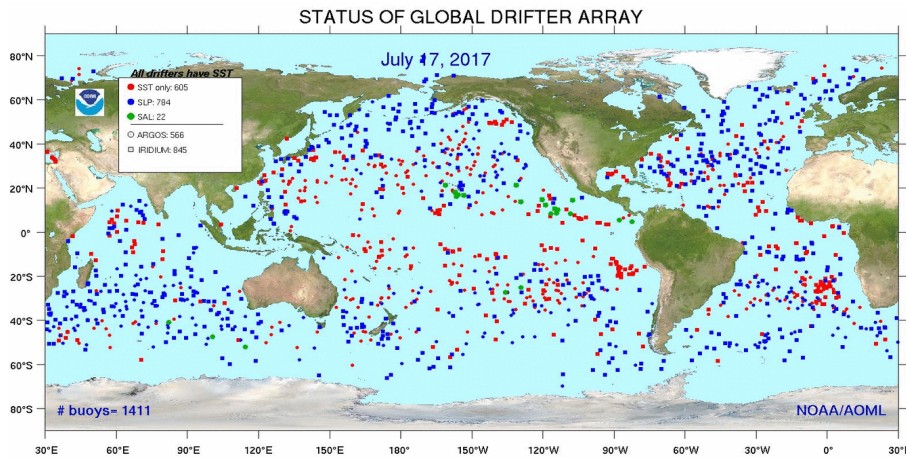

**Figure 1.** Current distribution of the global drifter array. Map regularly update by NOAA at http://www.aoml.noaa.gov/phod/dac/index.php. Colors indicate additional sensors carried by drifters.

The technologies to observe ocean currents have progressed in parallel to the own history of the ocean research. First measurements were already undertaken during the HMS *Challenger* expedition (1872-76). For several decades, the main source of information about the ocean currents had been ship-drift reports. Using about four million observations of ship-drift data, Richardson (1989) calculated annual and monthly mean surface currents in a $2° \times 5°$ grid. His charts served to identify

large gaps in the international databases, specially after the Second World War. Although mechanical current meters have been used since 1920s, their extended use by the oceanographic community started in the 1960s thanks to improved design, accuracy, and reliability of rotor-type current meters and the commercialization of modern acoustic Doppler currentmeters (Emery and Thomson, 2001). Simultaneously, attempts to infer deep ocean velocities by tracking drifting devices exploiting the Sound Fixing and Ranging (SOFAR) channel located around 1200 m depth (e.g. Rossby and Webb, 1970) were explored.

First prototypes designed independently by H. Stommel and J. Swallow in the 1950s (Swallow, 1955; Stommel, 1955) have now evolved into the RAFOS model allowing to unveil ocean currents in remote regions (Balwada et al., 2016). In the 1970s, the development of satellite positioning systems represented a remarkable advance that lead to setting up a global program for tracking Lagrangian drifters designed to follow the movement of surface waters (Lumpkin and Pazos, 2007). At present, Lagrangian drifters are able to provide hourly observations but with irregular coverage with approximately one point within a

5 degree box (Dohan and Maximenko, 2010, see figure 1).

The next major breakthrough was the launch of altimeter missions as Topex/Poseidon and ERS-1/2 in the early 1990s. Taking advantage of the precise measurements of sea level, global, near real-time maps of geostrophic velocities were derived for the first time at scales of several hundred kilometers and 5-10 days. Finally, it has been demonstrated that surface ocean currents can be directly measured using the Doppler effect, i.e. the frequency shift of an emitted electromagnetic wave due to

the relative motion between the emitter and the sea surface. This phenomenon is being exploited to retrieve current information from both satellite measurements provided by Synthetic Aperture Radar (SAR, see Chapron et al., 2005) and from coastal High

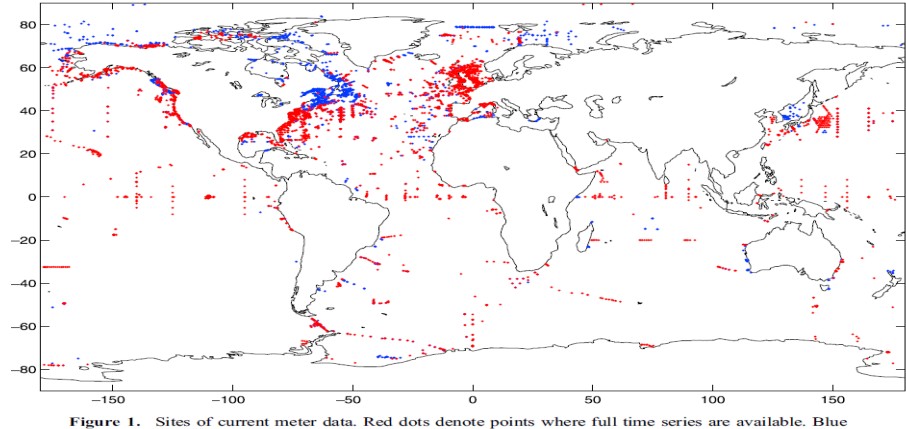

**Figure 1.** Sites of current meter data. Red dots denote points where full time series are available. Blue dots denote points where only temporal means and variances are available.

**Figure 2.** Summary of current observations from moorings and met-ocean buoys. Map available at Woods Hole Institution in http://www.whoi.edu/page.do?pid=68916. Colors indicated the availability of data, see a detailed explanation of data compilation in (Holloway, 2008)

Frequency (HF) radar stations (Paduan and Washburn, 2013). At the moment of writing this review, several missions able to measure the Doppler shift are under consideration by space agencies such as NASA and ESA. Some of these missions propose the use of altimeters (e.g SKIM) and scatterometers (e.g. DopSCAT) to this end; other missions, as SeaStar, are proposing new instruments.

Anticipating the goal of this review, today's ocean velocity observing system can be divided according to their regional extent: global and coastal.

At the global scale, the observations provided by mooring instruments are located mostly near and along the coasts, particularly in the northern hemisphere (Holloway, 2008; Scott et al., 2010, see figure 2). These moorings are usually clustered forming arrays of point-based currentmeters or current profilers from the ocean floor that provide limited temporal extent and
concentrated in the upper 100 m (Holloway et al., 2011). As a result, altimetry and Lagrangian drifters remain the sole source of information able to provide global coverage and have become the backbone of operative/operational synthesis products such as OSCAR (Bonjean and Lagerloef, 2002) and AVISO (CLS, 2016). However, the Rossby radius of deformation (providing the preferred horizontal scale of ocean structures) rapidly decreases from the equator to high latitudes (Stammer, 1997; Chelton et al., 1998). Then, the variability and interaction of currents with winds at mesoscale and submesoscale are not well captured
as today's observing systems fail to resolve horizontal gradients at these scales. In the case of SAR, some studies have already shown great potential in areas with very intense currents (Chapron et al., 2005; Rouault et al., 2010). The approach has two advantages: it is not affected by the presence of clouds and its high spatial resolution allows measurements close to the coast. There are, however, some limitations: only one component of the velocity is derived; the narrow swath limits the coverage; and the retrieved current speed may contain contributions other than the ocean current. Indeed, under a weak current regime the
dominant contribution is the wind-induced wave motion (Mouche et al., 2012).

No global simulations of the ocean circulation are assimilating ocean surface current observations. The main reason is the shortness of the records of direct retrievals of surface currents at global scale. As stated in the previous paragraph, long series of global surface current maps have been derived from altimeters, drifters, and surface winds. However, most of that information is already being directly assimilated (at a daily rate) in global simulations, providing constraining boundary conditions to the ocean circulation. As mesoscale is not well captured by these so-derived velocity maps, little improvement (if any) would be expected from their assimilation in global simulations. At regional scale, most of the assimilation efforts have focused on assimilating in-situ observations of currents derived from acoustic Doppler profiles and surface drifters. See, for example Carrier et al. (2014) and the references therein. On the context of remotely sensed velocity fields Santoki et al. (2013) were able to reduce the errors of the surface currents in a simulation of the Indian Ocean by assimilating five-day, $1° \times 1°$ OSCAR currents. More recently, Phillipson and Toumi (2017), found that adding OSCAR velocities in their assimilation scheme did not improve the forecasting skill obtained when drifters were assimilated alone. One of the reasons pointed out by the authors was the low frequency sampling (five days) of the OSCAR currents, together with the variable coverage of the satellite data used to derive OSCAR.

In coastal regions, the observation of surface currents has evolved differently because such an effort is driven by the need of risk assessment, environmental monitoring of marine protected areas and marine security. Together with in situ moored currentmeters, the use of HF radar systems in coastal areas has rapidly increased after the first decade of this century. Coastal HF radars have been shown to be able to resolve rapid changes. However, although the number of HF have been augmented, their coverage remains limited. Drifters can also be deployed in coastal zones, however their coverage remains sparse due to the elevated risk of beaching and/or equipment loss.

Contrarely to the case of global and regional assimilation experiments, a number of studies have been conducted to assess the advantages of assimilating remote sensed ocean currents in coastal simulations, as the number of coastal HF radars has increased in areas of strong economic activity.

As a kind of synthesis, the diagram in figure 3 illustrates how different components of the ocean observing system capture different parts of the range of processes associated with surface ocean currents. As such, a combination of direct measurements of surface currents by satellite and HF coastal radars is a promising approach to cope with both the resolution and fast dynamics characteristic of coastal areas and the mesoscale and slower evolution of surface currents in the open ocean regions. As stated before, direct measurements of surface currents by satellites remain quite limited. This situation has prompted to the development of various indirect methods, either by assuming dynamical constraints to SST images (Kelly, 1989; Vigan et al., 2000b; Chen et al., 2008) by applying pattern recognition techniques as neural networks (Côté and Tatnall, 1997) or the Maximum Cross Correlation technique (MCC Bowen et al., 2002; Afanasyev et al., 2002; Dransfeld et al., 2006). A better understanding of the dynamics in the upper layers of the ocean has allowed to propose a new framework based on the Surface Quasi-geostrophic equations (SQG Held et al., 1995; Lapeyre and Klein, 2006) able to retrieve sea surface currents from a single SST image (LaCasce and Mahadevan, 2006; Isern-Fontanet et al., 2006a; González-Haro and Isern-Fontanet, 2014). These methods open the way to develop techniques for direct assimilation of sea surface currents into general ocean forecasting systems, a question that, as commented above, has not yet impacted dynamic predictions, except for coastal radar applications.

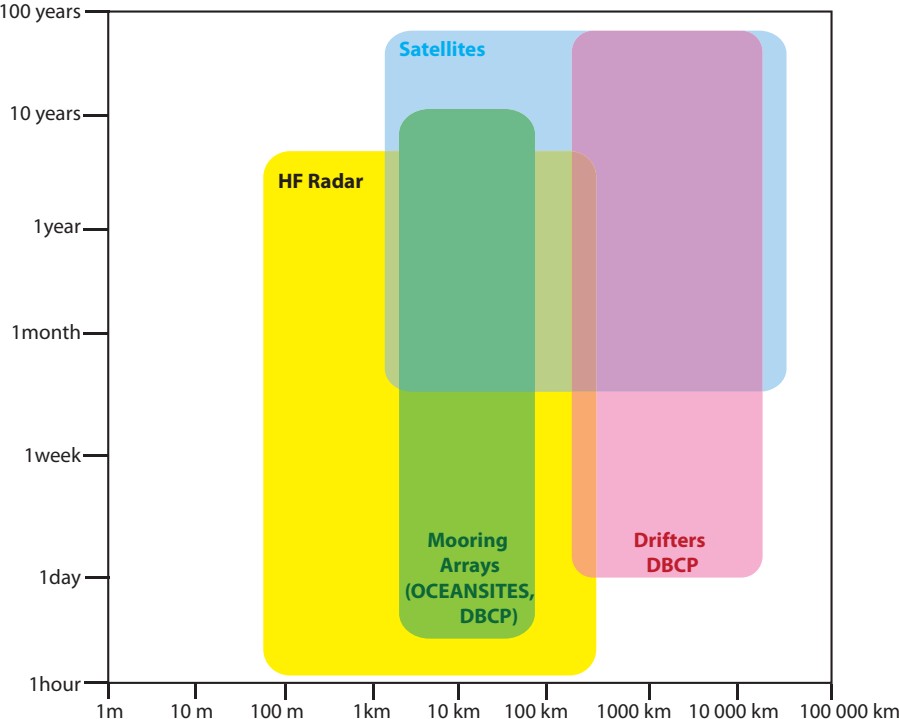

**Figure 3.** Spatio-temporal coverage by different technologies to measure sea surface currents. Adapted from the specifications sheet provided by the Global Ocane Observating System (GOOS), available at http://www.goosocean.org/components/com_oe/oe.php?task=download&id=34503&version=1.0&lang=1&format=1

The aim of this manuscript is to focus on reviewing two aspects of remote sensing of ocean surface currents. On the one hand, we are reviewing the different approaches that can be used to produce estimates of sea surface currents from remote sensing data (Sections 2 and 3). On the other hand, to review the advances in assimilation of sea surface currents, specifically centered on HF radar in coastal regions which is, up to now, the only source of remote sensing current measurements (Section 4). Is is expected that gained experience and the lessons learned from assimilating currents from HF radars can be translated, and applied, to global data assimilation systems if real-time, quasi-synoptic maps of ocean currents were available either from incoming satellite missions (e.g. SKIM, DopSCAT, SeaStar) or derived from the methods reviewed in section 2.

The structure of the manuscript is as follows. Section 2 provides an overview of the available approximations to retrieve currents from existing satellite observations. In particular, section 2.1 reviews the retrieval of geostrophic velocities from sea level measurements. Section 2.2 is devoted to analyze the complex upper layer dynamics, taking into account all the elements of the ocean-atmosphere interaction such as wind, waves. In section 2.3 we introduce the geometrical approaches used to infer sea surface velocity fields from the turbulent structure of the sea surface, as seen from multiple satellite sensors. Section 2.4 reviews the latest developments and the requirements to infer the sea surface velocity fields by inverting the potencial vorticity field applied to a single image. Section 3 focuses on the basic principles and sampling characteristics of coastal HF radars,

while section 4 reviews the attempts and limitations of the different assimilation techniques applied to HF radar observations: nudging, sequential and 4DVAR methods. Finally, section 5 provides a discussion about potential candidates to bridge the gap between global and coastal remote sensing of ocean currents.

## 2 Retrieval from satellite observations

At large scales Earth rotation dominates the dynamics of ocean currents. However, the inertia contribution will become increasingly important as the scales of the flow reduce or the the flow curvature grows. This motivates the introduction of the Rossby number:

$$\text{Ro} = \frac{U}{L f_0}, \tag{1}$$

where $U$, $L$ and $f_0$ represent the characteristic velocity, length scale and the Coriolis parameter respectively. Ro measures

the relative importance of the advective and the Coriolis terms in the momentum equation. At small Ro values, and without other sources of momentum such as wind and waves, the flow is close to the geostrophic balance implying an equilibrium between the Coriolis and pressure forces. Then, ocean currents can be simply derived from pressure measurements (or density, or sea surface height) invoking the geostrophic approximation. Agesotrophic contributions to ocean currents have two different sources: wind and waves on one side, and the departure from the geostrophic approximation due to larger values of Ro on the

other. It is worth mentioning that, although the geostrophic and ageostrophic contributions can conceptually be separated, any measurement of the ocean current is the result of all the contributions, making it difficult to assess the relative contribution of each one and the accuracy of the estimations.

### 2.1 Currents from Sea Surface Height

At zeroth order $\mathcal{O}(1)$ (i.e. Ro $<< 1$) and in absence of other sources of momentum (such as wind and waves) the horizontal

velocity field is non-divergent. As such, it is possible to define a stream function $\psi(\boldsymbol{x}, z)$ that only depends parametrically on the vertical coordinate $z$, such that the geostrophic velocity field $\boldsymbol{v}_0(\boldsymbol{x}, z)$ is given by (e.g. Vallis, 2006)

$$\boldsymbol{v}_0(\boldsymbol{x}, z) = \boldsymbol{e}_z \times \nabla_z \psi, \tag{2}$$

where $\boldsymbol{e}_z$ is the unit vector in the $z$ direction, $\boldsymbol{x} = (x, y)$ is the horizontal position and $\nabla_z \equiv (\partial_x, \partial_y, 0)$. This stream function is proportional to pressure at zeroth order, $p_0(\boldsymbol{x}, z)$:

$$\psi(\boldsymbol{x}, z) = \frac{1}{\rho_0 f_0} p_0 \tag{3}$$

with $\rho_0$ being a reference density. Close to the surface, the pressure field along an equipotential surface is related to the Sea Surface Height (SSH), $\eta(\boldsymbol{x})$, through the hydrostatic equation. Then, surface velocity at zeroth order becomes

$$\boldsymbol{v}_0(\boldsymbol{x}, 0) = \boldsymbol{e}_z \times \frac{g}{f_0} \nabla \eta. \tag{4}$$

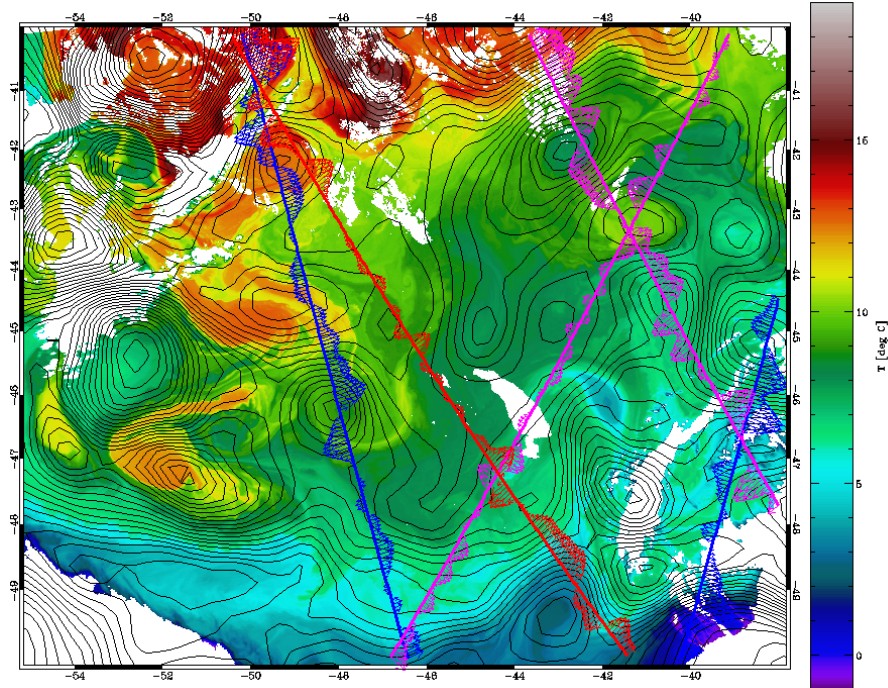

**Figure 4.** Sea Surface Temperature from MODIS Aqua with Sea Surface Height from AVISO (black lines) obtained from the combination of measurements provided by different altimeters. Lines show the available measurements in the period of $\pm$ 12 hours around the time the image was taken provided by Jason-1 (red), Envisat (blue) and GFO (purple). Arrows correspond to the cross-track geostrophic velocities.

This provides the fundamental framework that allows to retrieve surface ocean currents from the satellite measurements of SSH given by altimeters (see Robinson, 2004, for more details).

Current altimeters provide measurements of SSH along the satellite track with a sampling frequency of 20 Hz, implying a spatial resolution of the order of $\sim$300 m. The Power Spectral Density (PSD) of these measurements shows the presence of white noise (e.g. Le Traon et al., 2008; Xu and Fu, 2011, 2012; Dibarboure et al., 2014; Zhou et al., 2015), which is a major limiting factor for the estimation of ocean currents. Since noise has a stronger effect on small scales it is common to (low-pass) filter altimetric measurements before computing velocities. Nevertheless, this approach does not remove noise at large scales, which can be important in low energetic areas (e.g. Xu and Fu, 2012). Moreover, the level of noise strongly depends on the sea state, which makes it highly variable in space and time implying that the effective resolution of altimetric measurements is also variable. In a recent study, Dufau et al. (2016) have shown that the smallest scale that can be resolved by the new generation of altimeters is 40-50 km in areas of strong currents, but it can be as large as 90-100 km. This variability has motivated the development of adaptive approaches to better exploit the sampling capabilities of current altimeters (Isern-Fontanet et al., 2016a). During the last years there have been major improvements in radar altimetry technology that not only have reduced noise levels (Dufau et al., 2016) but also have reduced the impact of inhomogeneities in measurements (Dibarboure et al.,

2014). Nevertheless, current altimeters still present strong limitations in observing small scale features $\mathcal{O}(10\,\mathrm{km})$ not only due to noise but also due to temporal sampling (Chavanne and Klein, 2010). Finally, it is worth mentioning that current altimeters still have difficulties in providing measurements at distances between 10-50 km from the coast in spite of the advances done during the recent years (Cipollini et al., 2017).

Altimeter measurements only allow to retrieve the velocity perpendicular to the satellite track (equation 4). Two-dimensional fields are then typically obtained through the interpolation of measurements in space and time using classical Optimal Interpolation (OI) schemes (e.g. Le Traon et al., 1998). Figure 4 shows an example of the sampling capabilities of current altimeters and the performance of altimetric maps compared to a simultaneous thermal image. Altimeter measurements are available along satellite tracks (red, blue and purple straight lines) giving access to the cross-track velocity field (arrows). Black contours

correspond to the Absolute Dynamic Topography, i.e. the estimated heigh of the sea level to respect the geoid, obtained through the OI. The comparison with the thermal image unveils a mismatch in the location of the vortices, particularly in areas with no recent altimeter measurements such as the area around (46S, 52W). In this example it is also evident that small eddies seen in temperature measurements are not captured by current sea level data. This example ilustrates the two main problems of this technique. On the one hand, the separation between tracks and the time sampling reduce the spatial resolution in comparison

with the one achieved by the along-track measurements. Chelton et al. (2011) estimated that the shortest wavelength that can be achieved by the interpolated two-dimensional fields is $\lambda \sim$150-200 km, implying that vortices with diameters smaller than 75 - 100 km cannot be observed by altimeters. This gives rise to the so-called altimetric gap, i.e. the range of scales that cannot be currently observed by altimeters. On the other hand, the limited amount of altimeters as well as the rapid evolution of some structures may induce errors in the location and geometry of ocean vortices. Pascual et al. (2006) showed that the difference

between using two or four altimeters induces RMS difference in Sea Level Anomalies up to 10 cm, differences in the Eddy Kinetic Energy as big as $400\,\mathrm{cm^2 s^{-2}}$, and comparison against drifting buoys unveiled important errors in the location of some vortices (see figure 3 in Pascual et al., 2006). Moreover, Isern-Fontanet (2016) and Isern-Fontanet et al. (2017b) have shown that SSH maps derived from altimetry does not capture some fast evolving patterns seen in SST at the Alboran Sea.

Several efforts have been done during the last years to improve the ability to obtain two-dimensional velocities from along-

track data. For example, Ubelmann et al. (2015) have recently proposed a new approach to interpolate the sparse altimetric measurements into a regular grid based on the advection of Potential Vorticity (see section 2.4 below) during short periods of time ($< 20$ days) of scales smaller than $\sim 300$ km. This method has been recently adapted to the interpolation of along-track altimetric measurements improving the performance of the classical OI schemes (Ubelmann et al., 2016). Other proposed approaches attempt to improve altimetric maps using a two-step approach. That is, after the standard maps are computed,

the residuals to respect along track data are reinterpolated using different correlation functions that may include bathymetric constrains (see Escudier et al., 2013, and references therein). It is expected that, in the following years, the two-dimensional SSH field will be directly measured by novel satellite missions like the Surface Water and Ocean Topography (SWOT) mission using swath altimetry (Durand et al., 2010).

Another approach aiming to improve the direction of currents derived from altimetric measurements is based on the use

of complementary satellite observations such as those obtained from thermal and visible measurements. Measurements of sea

surface temperature, particularly those from infrared observations, are very precise in locating ocean structures such as as fronts. Strong fronts have a tendency to be aligned with currents. This allows to retrieve two-dimensional velocity fields associated to thermal fronts, or even chlorophyll concentration patterns. In particular, given the cross-track geostrophic velocity $v_\perp(x)$, the along-track component $v_\parallel(x)$ can be estimated as

$$v_\parallel(\boldsymbol{x}) = v_\perp(\boldsymbol{x}) \tan \alpha_f, \tag{5}$$

where $\alpha_f$ is the angle between the front and the vector orthogonal to the altimetric track. This approach has some drawbacks: it is sensitive to noise, and it is only valid for strong fronts becoming a region-dependent approximation (GlobCurrent, 2017). The underlying idea can also be pushed to correct two-dimensional altimetric maps. As before, under the assumption that strong fronts are a proxy of the geostrophic stream-lines, the information is propagated along-fronts using a Lagrangian framework and the altimetric velocities are corrected in both, the direction using the orientation of the front and the speed using the variation of intensity of the thermal gradient (GlobCurrent, 2017) .

The advective term, i.e. $\boldsymbol{v} \cdot \nabla \boldsymbol{v}$, in the momentum equation is absent in the geostrophic approximation because is $\mathcal{O}(\mathrm{Ro})$ in the expansion in terms of $\mathrm{Ro}$ (Vallis, 2006). If the flow is considered to be axisymmetric, $\boldsymbol{v}(r) = v_\theta \boldsymbol{e}_\theta$, the advection term becomes $-r^{-1} v_\theta^2 \boldsymbol{e}_r$, where $r$ is the radius of curvature and $\boldsymbol{e}_r$ and $\boldsymbol{e}_\theta$ are the radial and tangential unit vectors. Momentum equations can be then easily solved giving rise to the Gradient Wind solution (e.g. Holton, 1992). This provides a first correction to the geostrophic velocities derived from altimetry, which can be up to 50% of the geostrophic velocity in intense vortices (Penven et al., 2014). This corrrection, which depends on the curvature of the streamlines, can be implemented using the iterative method proposed by Endlich (1961) and Arnason et al. (1962):

$$\boldsymbol{v}^{n+1}(\boldsymbol{x}) = \boldsymbol{v}_0 + f_0^{-1} \boldsymbol{e}_z \times (\boldsymbol{v}^n \cdot \nabla_z \boldsymbol{v}^n) \tag{6}$$

The iterations stop once the velocity increment falls below a threshold or it starts to increase (Penven et al., 2014).

## 2.2 Currents from wind and waves

Altimeter-derived geostrophic currents only account for a part of the surface circulation. The ocean response to atmospheric forcing (the most relevant component of the surface current) must be added to the geostrophic currents. The launch of scatterometers has allowed to measure several parameters characterizing the processes in the ocean-atmosphere interface (wind stress, roughness, wave height, etc) enabelling to quantify the wind-driven components of the sea surface currents. To understand and review the recent efforts to include atmosphere-ocean processes in retrieving the sea surface currents we start from the classical approach by W. Ekman (Ekman, 1905) who solved the momentum equations for a semi-infinite ocean forced by wind (section 2.2.2). However, his solution didn't include the contribution from waves, which were added much later (section 2.2.3). Both solutions solve the momentum equations for a steady, hydrostatic and Boussinesq flow while, recent approaches generalized the problem by writing the momentum equations in terms of the turbulent stress (section 2.2.1).

### 2.2.1 Momentum equations

The momentum equations for a steady, hydrostatic and Boussinesq flow are given by

$$f\boldsymbol{e}_z \times (\boldsymbol{v} + \boldsymbol{v}_S) = \frac{1}{\rho_0}\nabla p + \frac{1}{\rho_0}\frac{\partial \boldsymbol{\tau}}{\partial z} + b\boldsymbol{e}_z, \tag{7}$$

where $\boldsymbol{v}(\boldsymbol{x},z) = (u,v)$ is the total horizontal velocity field, $\boldsymbol{\tau}(\boldsymbol{x},z) = (\tau_x,\tau_y)$ the turbulent stress, $b(\boldsymbol{x},z) = -g\rho/\rho_0$ is buoyancy and $p(\boldsymbol{x},z)$ and $\rho(\boldsymbol{x},z)$ a perturbation pressure and a perturbation density with respect to the reference density $\rho_0$, such that $|\rho| \ll \rho_0$ and which has associated a reference pressure given by $\partial_z p_0 = -g\rho_0$, and $g$ is gravity. Contrary to the standard formulation of the Boussinesq flow (e.g. Vallis, 2006), equation 7 contains the non-linear contribution from waves: the Stokes drift $\boldsymbol{v}_S(\boldsymbol{x},z) = (u_S,v_S)$ which is the Lagrangian mean velocity due to waves. Notice that, in the context of wave-driven currents, $\boldsymbol{v}(\boldsymbol{x},z)$ is the quasi-Eulerian velocity defined as the Lagrangian mean velocity over a wave period minus the Stokes drift (e.g. Polton et al., 2005). As in section 2.1, the Rossby number is assumed to be small allowing to neglect non-linear terms from the equation. Vertical boundary conditions are

$$\boldsymbol{\tau}(\boldsymbol{x},0) = \boldsymbol{\tau}_w \tag{8}$$

$$\boldsymbol{\tau}(\boldsymbol{x},-H) = 0, \tag{9}$$

with $\boldsymbol{\tau}_w$ being the surface wind stress and $z = -H$ the no-stress depth. Turbulent stress is commonly parametrized as a simple gradient transfer eddy-viscosity model

$$\boldsymbol{\tau}(\boldsymbol{x},z) \equiv \rho_0 A_v \frac{\partial \boldsymbol{v}}{\partial z} \tag{10}$$

with $A_v(z)$ being the eddy viscosity (e.g. Polton et al., 2005; Cronin and Kessler, 2009; Wenegrat and McPhaden, 2016). It is common to rewrite equation 7 in its complex form as

$$if(\tilde{V} + \tilde{V}_S) = -\frac{1}{\rho_0}\tilde{\nabla}p + \frac{1}{\rho_0}\frac{\partial \tilde{\tau}}{\partial z} \tag{11}$$

$$0 = -\frac{1}{\rho_0}\frac{\partial p}{\partial z} + b, \tag{12}$$

with $\tilde{V}(\boldsymbol{x},z) = u + iv$, $\tilde{V}_S(\boldsymbol{x},z) = u_S + iv_S$, $\tilde{\tau}(\boldsymbol{x},z) = \tau_x + i\tau_y$ and $\tilde{\nabla} = \partial_x + i\partial_y$.

At the ocean surface, $\tilde{V}(\boldsymbol{x})$ can be obtained from equations 11 and 12 using satellite observations. The perturbation pressure at the ocean surface can be derived from altimetric measurement of SSH through $p(\boldsymbol{x}) = \rho_0 g\eta$. The buoyancy can be expressed in terms of $T_s(\boldsymbol{x})$ and $S_s(\boldsymbol{x})$:

$$b_s(\boldsymbol{x}) = -\frac{g}{\rho_0}\left[\alpha_T(T_s(\boldsymbol{x}) - T_0) + \beta_S(S_s(\boldsymbol{x}) - S_0)\right], \tag{13}$$

using SST from infrared and microwave radiometers and SSS from microwave radiometers as well. Here, $\alpha_T$ is the thermal expansion coefficient and $\beta_S$ is the haline contraction coefficient. Finally, the wind stress term $\boldsymbol{\tau}_w$ can be derived from scatterometer measurements. This approach is used to generate ocean current products such as OSCAR (Lagerloef et al., 1999;

Bonjean and Lagerloef, 2002; Johnson et al., 2007) and GEKCO (Sudre and Morrow, 2008; Sudre et al., 2013), without including the Stokes drift ($\tilde{V}_S$ term in equation 11).

The Stokes drift contribution is difficult to be retrieved from satellite observations. As it has been seen above, it can be defined as the difference between the Eulerian and Lagrangian velocities due to wave motion averaged over a wave period. In the case of a monochromatic wave, the Stokes drift can be computes as (Phillips, 1977)

$$\tilde{V}_S = V_S \exp(2k_w z)\tilde{e}_k, \ V_S = a_w^2 \sigma_w k_w, \tag{14}$$

where $a_w$ is the wave amplitude, $\tilde{e}_k$ the direction of propagation in complex notation, $k_w$ is the wavenumber and $\sigma_w$ the wave frequency. This equation is unrealistic for the real ocean, where the wave field is the result of the combination of many modes. It is therefore necessary to have information of wave statistics. In particular, the Stokes drift is proportional to the third moment of the wave spectrum (e.g. Ardhuin et al., 2009). Some necessary information about surface waves can be retrieved from altimeters which also provide information about the Mean Square Slope (MSS) and the Significant Wave Height (SWH). From these values it is possible to empirically retrieve the mean period (Gommenginger et al., 2003). These measurements, however, do not provide the direction of propagation. Such information can be retrieved from Synthetic Aperture Radars, which provide information on the directional wave spectra although for wavelengths longer than 150 m. The Stokes drift can also be directly obtained from fields observable from satellites such as wind (e.g. scatterometers) and wave height (e.g. altimeters) using empirical models like the one used by Ardhuin et al. (2009). Nevertheless, the Stokes drift is in practice estimated using wave parameters provided by wave models (e.g. Hui and Xu, 2016), although these estimates may vary widely with model parameterizations (Ardhuin et al., 2009; Rascle and Ardhuin, 2013).

Since the momentum balance of equations 11 and 12 is linear and, assuming that pressure gradients are not related to local wind nor waves, they are often separated into a geostrophic velocity field $\tilde{V}_g$, which depends on the pressure gradients and can be derived from SSH measurements (equations 2 and 4), and an ageostrophic field $\tilde{V}_a$ driven by wind and waves.

Besides, the parametrization of turbulent stress in terms of the velocity field allows combining equations 7 and 10, resulting in a second-order linear equation for the velocity. However, an alternative approach is obtained differentiating equation 7 and manipulate it to obtain an equation for the turbulent stress $\tau(\boldsymbol{x}, z)$ known as the Generalized Ekman Model (Bonjean and Lagerloef, 2002; Cronin and Kessler, 2009; Wenegrat and McPhaden, 2016) or the Turbulent Thermal Wind Balance (Gula et al., 2014; McWilliams et al., 2015):

$$A_v \frac{\partial^2 \tilde{\tau}}{\partial z^2} - if\tilde{\tau} = \rho_0 A_v \tilde{\nabla} b - \rho_0 A_v if \frac{\partial \tilde{V}_s}{\partial z}. \tag{15}$$

Once stress has been retrieved, velocity can be computed using equation 11. This is the approach used by the OSCAR product without including the Coriolis-Stokes term. This approach improves the solution of Lagerloef et al. (1999) and has been extensively validated (e.g. Bonjean and Lagerloef, 2002; Johnson et al., 2007). Recently, Wenegrat and McPhaden (2016) have

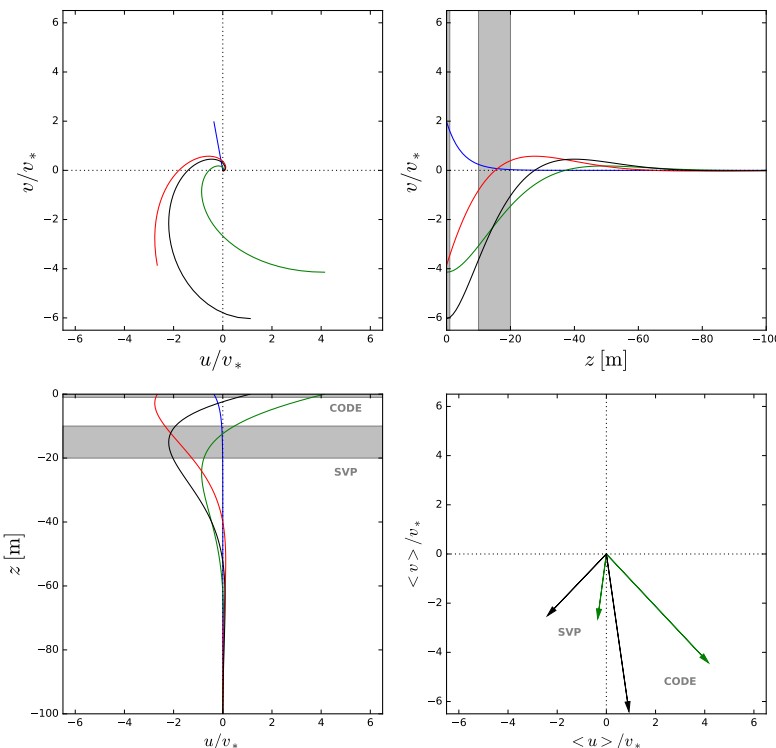

**Figure 5.** Ageostrophic velocity field for the Ekman component (green), the 'Eulerian' Stokes component (blue), the Ekman-Stokes component (red) and the resulting velocity (black). The parameters used are the same as in Polton et al. (2005). Wind and wave propagation is in the x-direction. All velocities are normalized by the friction velocity $v_*$. The paremeters used are the same as in Polton et al. (2005). Arrows in the lower-right plot correspond to the total (black) and Ekman (Green) transport a SVP and a CODE drifter would see obtained by integrating velocities for the layers marked with gray bands.

provided and approximate general solution to this equation based on Green's function given by

$$\tilde{\tau}(\boldsymbol{x}, z) = \tilde{\tau}_w \left[ \frac{A_v(z)}{A_v(0)} \right]^{\frac{1}{4}} \frac{\sinh[\xi(z)]}{\sinh[\xi(0)]} \tag{16}$$

$$+ \rho_0 \int_{-h}^{0} G(z, s) \left[ \tilde{\nabla} b + i f \frac{\partial \tilde{V}_s}{\partial z} \right] ds, \tag{17}$$

where

$$\xi(z) = \sqrt{if} \int_{-h}^{z} A_v(z')^{-\frac{1}{2}} dz' \tag{18}$$

and $G(z,s)$ is the Green's function given by equation 9 in Wenegrat and McPhaden (2016). This solution is quite general and admits different parameterizations of turbulent viscosity coefficient $A_v(z)$. In addition, this solution can also include the forcing from buoyancy and the effect of waves.

Finally, it is worth mentioning that the use of forcing data (SST and SSH) with different effective resolutions in equation 7 may induce unphysical imbalance associated to the different spatial resolution of products such as SST (of the order of 1 km for IR radiometers) and SSH (of the order of 50-100 km for altimetric maps). Consequently, the spatial resolution of this approach is limited by the field with the lowest effective resolution. A possible approach to increase the spatial resolution of altimetric maps (see the discussion in section 2.1) consists in merging altimetric maps with Lagrangian measurements. Indeed, Taillandier et al. (2006) proposed a variational algorithm that has been successfully used by Berta et al. (2015) to combine CODE data and altimetric maps, who found that not only it is possible to restore some of the variability missed in altimetric maps but also ageostrophic contributions beyond the simple Ekman model. Obviously, this approach is limited by the availability of enough drifter data.

### 2.2.2 Wind solution

Ekman (1905) provided a solution to the ageostrophic part of equation 11 by setting $\tilde{V}_S = 0$, $A_v(z) = A_0$, where $A_0$ is a constant, and modifying the bottom boundary condition (equation 9) by

$$\boldsymbol{u} \to 0 \text{ and } \boldsymbol{\tau} \to 0 \text{ as } z \to -\infty. \tag{19}$$

This solution only depends on the wind stress and the constant value given to $A_0$,

$$\tilde{V}_a(\boldsymbol{x}, z) = \tilde{V}_E (1 - i) \exp\left(\frac{z(i+1)}{d_E}\right), \tag{20}$$

where

$$\tilde{V}_E(\boldsymbol{x}) = \frac{\tilde{\tau}_w}{\rho_0 \sqrt{2f A_v}}, \tag{21}$$

and $d_E = \sqrt{2A_v f^{-1}}$ is the Ekman depth (see figure 5). If turbulent stress (equation 10) is assumed to be a linear function of depth, i.e.

$$\boldsymbol{\tau}(\boldsymbol{x}, z) \equiv \frac{\boldsymbol{\tau}_w}{H} z + \boldsymbol{\tau}_w, \tag{22}$$

the resulting ageostrophic velocities are given by

$$\tilde{V}_a(\boldsymbol{x}, z) = -i \frac{\tilde{\tau}_w}{\rho_0 f H}, \tag{23}$$

which is the so-called slab model characterized by a vertically homogeneous ageostrophic velocity field.

Both solutions depend on $\boldsymbol{\tau}_w(\boldsymbol{x})$, which can be retrieved from satellite measurements, and some parameters, i.e. $A_0$ and $H$, that have to be determined. Notice, however, the key differences between these two solutions. Ekman solution has the ageostrophic velocity field that decrease with depth and surface velocities are at $\frac{\pi}{4}$ rad to the right (left) of wind in the Northern (Southern) Hemisphere while, in the slab model solution, velocities are vertically homogeneous in the upper layer and surface velocity is at $\frac{\pi}{2}$ rad to the right (left) of wind in the Northern (Southern) Hemisphere. The main approaches to retrieve the wind-induced currents usually do not attempt to reconstruct the vertical profile of velocities but focus on determining the average motion of a layer and may take into account the singularity at the equator due to the Coriolis parameter (e.g. Lagerloef et al., 1999). Notice that other parameterizations of turbulent shear, e.g. through the dependence of the eddy viscosity on wind stress or shear of turbulence fluxes, are possible (see Wenegrat et al., 2014, and references therin).

Rather than using the theoretical models given by equations 20 and 23, some approaches to determine the wind-induced ageostrophic contribution of the velocity field are physically-based statistical models calibrated with independent observations of the velocity field, typically surface drifters (e.g. Lagerloef et al., 1999; Rio and Hernandez, 2003; Poulain et al., 2012) . The most widely used model assumes

$$\tilde{V}_a(\boldsymbol{x}) \equiv B\tilde{\tau}_w \exp(i\theta), \tag{24}$$

where $B$ and $\theta$ are constants to be fitted with observed velocities (Ralph and Niiler, 1999; Rio and Hernandez, 2003; Poulain et al., 2009; Chiswell, 2016). As a consequence, the resulting velocities derived from satellite wind measurements will be representative of the motion at the depth of measurements. The angle $\theta$ can be derived from the observation of drifter trajectories. It has been found to be within the range $20°$-$60°$ for the global ocean and the Eastern Mediterranean sea (Rio and Hernandez, 2003; Poulain et al., 2009) using SVP drifter trajectories that represent the motion of a 10 m layer centered at 15 m deep (see Lumpkin et al., 2017, and references therein). Moreover, Poulain et al. (2009) found very small differences in the direction between the SVP and CODE (drogued at $\sim$ 1m Lumpkin et al., 2017) buoys in the Mediterranean sea when fitting the model given by equation 23. On the contrary, Rio et al. (2014) found large differences between angles using SVP and Argo drifters with a geographical and seasonal dependence.

### 2.2.3 Wave solution

The interaction of the Stokes drift with planetary vorticity introduces and additional force on the momentum equations known as the Coriolis-Stokes force. As a consequence, the ideal solution of the ageostrophic component of the velocity has additional

terms to respect equation 20 given by (Polton et al., 2005)

$$\tilde{V}_a(\boldsymbol{x}, z) = \tilde{V}_E(1-i)\exp\left(\frac{z(i+1)}{d_E}\right) \tag{25}$$

$$+ \frac{\tilde{V}_S d_S}{d_E}(1-i)\exp\left(\frac{z(i+1)}{d_E}\right)\frac{\frac{d_E^2}{2d_S^2}}{\left(1+i\frac{d_E^2}{2d_S^2}\right)} \tag{26}$$

$$- \frac{\tilde{V}_S}{\left(1+i\frac{d_E^2}{2d_S^2}\right)}\exp\left(\frac{z}{d_S}\right) \tag{27}$$

assuming the same boundary conditions as in the classical Ekman solution (equation 19). Here, $\tilde{V}_E$ is the Ekman current at the ocean surface (equation 21), $\tilde{V}_S$ the stokes velocity and $d_S = 1/(2k_w)$, where $k_w$ is the wavevector (see equation 14). The Coriolis-Stokes forcing changes the direction of the ageostrophic component. It also has an exponentially decaying vertical contribution that could be of the same extent as the Ekman term. Therefore, the heuristic model given by equation 24, when fitted to wind measurements and drifter trajectories, might mix the wind and the wave contributions.

Figure 5 plots the ideal solutions given by equation 27. It shows the total solution (black) decomposed into the three solution discussed above: Ekman (green), 'Eulerian' Stokes (blue) and Ekman-Stokes (red) as well as the integration of these solutions for the depths of the CODE and SVP drogues. The values used are the same as in Polton et al. (2005). As it is evident in the figure these drifters are expected to have different direction in comparison with SVP drifters. Although the determination of upper wind and wave-driven currents provided by the above equation may not be accurate (see for example Rascle and

Ardhuin, 2009), observations do see, in general, differences between different types of drifters (Rio et al., 2014). Interestingly, these differences are very small in the Mediterranean (Poulain et al., 2009, 2012). Although the slab model has vertically homogeneous velocities, the inclusion of the Coriolis-Stokes induces vertical variations of the velocity field since, in general $d_S$ is smaller than the Mixed Layer Depth ($H$ in equation 23). In a recent paper Hui and Xu (2016) have included the Stokes-Coriolis force into the model proposed by Lagerloef et al. (1999) showing an improvement of the velocity field observed by

SVP drifters to respect the standard OSCAR products, particularly in the Southern Ocean. The use of a monochromatic profile (equation 14), however, leads to an underestimation of the near-surface shear and an overestimation of the deep Stokes drift (e.g. Ardhuin et al., 2009) which has lead Breivik et al. (2016) to propose an improved Stokes drift velocity profile based on the Phillips spectrum.

## 2.3   Currents from a sequence of tracer images

The apparent motion of surface tracers such as SST and chlorophyll concentration suggests the use of sequences of satellite images to retrieve the velocity field that originated this motion. This is being done using two main approaches: feature tracking and inverting the conservation equation for the tracer, which, in general is given by

$$\frac{\partial c}{\partial t} + \boldsymbol{v} \cdot \nabla_z c = \dot{C}, \tag{28}$$

where $c(\boldsymbol{x}, t)$ can be SST or chlorophyll concentration or even the MSS and $\dot{C}$ are the sources and sinks of this tracer, including

the vertical advection contribution, i.e. $-w\partial_z c$, where $w$ is the vertical velocity component. It is important to realize that

the advection term $\boldsymbol{v} \cdot \nabla_z C$ is the inner product between velocity and tracer gradients, which implies that only the velocity component parallel to tracer gradient can be retrieved by inverting equation 28. This is what is known as the aperture problem. However, while the wealth of satellite measurements of SST points to their use for estimating ocean currents although, this approach is not necessarily the best choice in certain situations. The skin depth of SST is of the order of a few $\mu$m implying that air-sea interactions can mask the presence of oceanic structures. Moreover, the algorithms used to retrieve SST introduce additional noise. Therefore, in some situations Brightness Temperature (BT) is better suited than SST for the estimation of currents (e.g. Bowen et al., 2002; Isern-Fontanet and Hascoët, 2014). Notice, however, that BT does not contain the atmospheric correction, implying that temperatures are lower and atmospheric patterns may contaminate the image. With respect to the chlorophyll concentration, it has the advantage of integrating information of the upper tens of meters, so it is able to outline ocean patterns better than SST. Nevertheless, less images are available since ocean color data can only be used during daytime and chlorophyll amount is not conservative (even on daily cycle). Interestingly, Warren et al. (2016) have shown that slightly better performance can be obtained using the individual visible channels (in the blue-green end of the spectrum); similarly to the use of BT instead of SST. In any case, the use of ocean color and SST data are limited by the need of having cloud-free sequences of images.

The standard approach used in feature tracking is the so called Maximum Cross-Correlation method (Emery et al., 1986; Bowen et al., 2002; Barton, 2002). The underlying idea is quite simple: given a template of $N_x \times N_y$ grid points in an image at time $t_0$, it consists in searching which sub-window of size $N_x \times N_y$ has the maximum cross-correlation within a larger search window in an image at time $t_0 + \Delta t$ and take the displacement vector between images as the velocity field. This approach has been mainly applied to SST (e.g. Dransfeld et al., 2006; Castellanos et al., 2013; Doronzo et al., 2015) although recently it has been also applied successfully to ocean color data (e.g. Yang et al., 2015; Warren et al., 2016; Hu et al., 2017).

An alternative approach consists on tracking the biogenic surface slicks. These are slicks formed by monomolecular slicks that modify the surface tension and therefore affect capillary waves reducing the backscatter or microwave radar emissions. This allows to observe such slicks in MSS images provided by SAR and use the MCC technique to retrieve currents. This approach was successfully tested by Qazi et al. (2014), who used SAR data from Envisat and ERS-2 separated by only 30 minutes. Although the use of SAR data allows to overcome the limitation imposed by cloud-coverage, the interpretation of MSS is strongly dependent on weather conditions (Robinson, 2004; Kudryavtsev et al., 2005) implying that it can only be applied for winds within the range 2-7 m/s (Qazi et al., 2014). Marcello et al. (2008) proposed to improve the MCC approach using a two-step procedure: in the first step, image segmentation is used to unveil the patterns present in the image, which are tracked in the second step. This tracking combines MCC vectors and Optical Flow methods, i.e. inversion of equation 28 with $\dot{C} = 0$. In general, the resulting velocity field is sparse and is post-processed to retrieve a smoother field (e.g. Afanasyev et al., 2002) or it is combined with altimetric measurements (e.g. Abraham, 1998; Wilkin et al., 2002). Notice that, the MCC approach requires high resolution data such as the observations provided by infrared and visible radiometers (resolutions $\sim 1$ km) but the resulting velocity field has spatial resolutions of the order of the window used to track features ($\sim 20$ km, e.g. Bowen et al., 2002).

An alternative to feature tracking is to solve the heat equation, which provides an equation for the evolution of SST. Integrating over the Mixed Layer (ML), the heat equation can be written as

$$\frac{\partial T}{\partial t} + \boldsymbol{v} \cdot \nabla_z T = \kappa \nabla^2 T + \frac{Q}{\rho_0} - w_e \frac{T - T_d}{H}, \tag{29}$$

where $Q(\boldsymbol{x}, t)$ are the heat fluxes, $\kappa$ is the thermal diffusion, $w_e$ is the entrainment velocity at the base of the ML which is non-zero only if there is a deepening of the ML (e.g. see Klein and Hua, 1990) and $T_d$ is the temperature below the ML. In the ocean, the Péclet number is smaller than one, implying that the diffusion term can be removed from equation 29. As outlined above, only the cross-isotherm component of the velocity can be retrieved unless additional constrains are taken into account. To solve this problem, Kelly (1989) and Kelly and Strub (1992) used horizontal divergence $\nabla_z \cdot \boldsymbol{v}$ and the vertical component of vorticity $(\nabla \times \boldsymbol{v})_z$ as regularizing constrains for the cost function given by

$$\mathcal{L}(u, v) = \left[ \frac{\partial T}{\partial t} + \boldsymbol{v} \cdot \nabla_z T - \dot{\mathcal{T}} \right]^2 + a^2 \left[ \nabla \cdot \boldsymbol{v} \right] + b^2 \left[ \nabla \times \boldsymbol{v} \right]_z, \tag{30}$$

with $\dot{\mathcal{T}}(\boldsymbol{x}, t)$ being the source terms in equation 29 and $a$ and $b$ two penalty parameters to tune the influence of vorticity and divergence, which has been solved using a wide variety of numerical schemes (Kelly, 1989; Vigan et al., 2000a; Chen et al., 2008). An alternative approach to solve the aperture problem consists on using background velocity information (Piterbarg, 2009) such as altimetry (Rio et al., 2016). In that case, the velocity field is given by

$$\boldsymbol{v}(\boldsymbol{x}) = \boldsymbol{v}_{alt} - \frac{\nabla_z T \cdot \left[ \partial_t T + \boldsymbol{v}_{alt} \cdot \nabla_z T - \dot{\mathcal{T}} \right]}{(\nabla_z T)^2}, \tag{31}$$

where $\boldsymbol{v}_{alt}(\boldsymbol{x})$ is the velocity field given by altimeters. This methodology has the same problems than MCC as it requires a sequence of cloud-free images. Nevertheless, since it does not attempt to track features it could be applied to low resolution SST data such as the measurements provided by microwave radiometers, which are not affected by clouds. As such, its use might help to improve the topology of SSH fields if not enough altimeters are available (see discussion in section 2.1).

The need to solve the differential equation 29 imposes constrains on the spatial resolution $\Delta x$, which is controlled by the spacing between satellite images $\Delta t$ and the cross-isotherm velocity $U_T$ (Kelly, 1989), i.e.

$$\Delta x > U_T \Delta t. \tag{32}$$

Taking $U_T \approx 16$ km/day and $\Delta t \approx 1$ day gives $\Delta x > 16$ km, while $\Delta t \approx 6$ hours implies $\Delta x > 4$ km. If altimetric maps are used to solve the aperture problem then, the effective spatial resolution will be reduced to that of altimetry (see section 2.1).

## 2.4 Currents from a single tracer image

The methods decribed in sections 2.1-2.3 rely on altimetric measurements to obtain the topology component of the velocity field. As it was discussed in section 2.1, altimeters are limited by current technology (noise level, distance to coast) and sampling geometry (difficulty to retrieve two-dimensional currents). This fact has motivated the development of alternative approaches that exploit the characteristics of SST measurements.

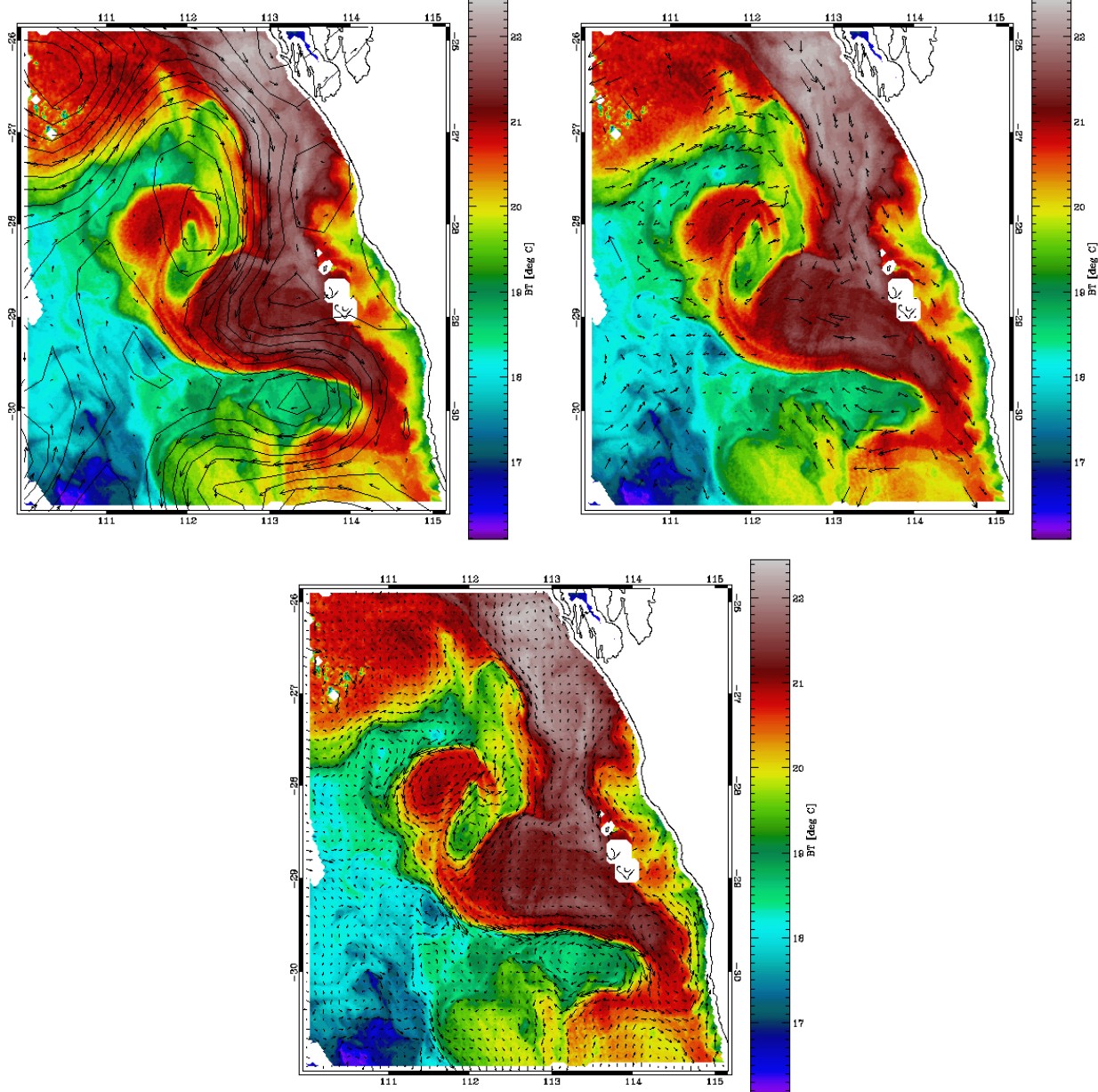

**Figure 6.** Sea Surface Temperature from MODIS Aqua with Sea Surface Height from AVISO (black lines) obtained from the combination of measurements provided by different altimeters. Lines show the available measurements in the period of $\pm$ 12 hours around the time the image was taken provided by Jason-1 (red), Envisat (blue) and GFO (purple). Arrows correspond to the cross-track geostrophic velocities.

The necessary framework can be found at $\mathcal{O}(\text{Ro})$ in the so-called Quasi-Geostrophic approximation (Vallis, 2006). Within this framework, Potential Vorticity (PV) anomaly $q(\boldsymbol{x}, z)$ is related to the geostrophic stream function (equation 3) through

$$\nabla_z^2 \psi + \frac{\partial}{\partial z}\left( \frac{f_0^2}{N^2} \frac{\partial \psi}{\partial z} \right) = q, \tag{33}$$

where is $N(z)$ the Brunt-Väisälä frequency. The hydrostatic equation provides the appropriate boundary conditions at the ocean surface:

$$f_0 \left.\frac{\partial \psi}{\partial z}\right|_s = b_s, \tag{34}$$

where $b_s(\boldsymbol{x})$ is the sea surface buoyancy (SSB), and at the ocean bottom ($z = -H$)

$$\left.\frac{\partial \psi}{\partial z}\right|_H = 0. \tag{35}$$

Alternativelly,

$$\lim_{z \to -\infty} \frac{\partial \psi}{\partial z} = 0, \tag{36}$$

where it is assumed that the bottom is far enough. Then, using the principle of invertibility of PV (Hoskins et al., 1985), the geostrophic stream function can be computed from the knowledge of surface buoyancy, that can be retrieved from SST and SSS measurements (equation 13); $N(z)$ that can be obtained from climatologies or density profiles from Argo buoys and the knowledge of PV. Unfortunately, PV is not known and cannot be derived from satellite measurements. Nevertheless, Lapeyre and Klein (2006) showed that the large-scale forcing in density and PV can lead to the property that the interior PV mesoscale anomalies are correlated to the surface buoyancy anomalies in the upper ocean. In that case, the PV anomaly can be separated as

$$q(\boldsymbol{x}, z) \approx \xi(z) b_s(\boldsymbol{x}), \tag{37}$$

with $\xi(z)$ being a function that specifies the amplitude of PV anomaly. Equation 33 can be used to retrieve the stream-function from surface buoyancy, i.e. from SST and SSS measurements.

Bretherton (1966) and Lapeyre and Klein (2006) proposed to solve this problem by splitting it into two solutions:

$$\psi(\boldsymbol{x}, z) = \psi_{srf} + \psi_{int}. \tag{38}$$

That is, as the sum of a surface solution $\psi_{srf}(\boldsymbol{x}, z)$, obtained assuming non-zero surface buoyancy and zero interior PV ($b_s \neq 0$ and $q = 0$), and an interior solution $\psi_{int}(\boldsymbol{x}, z)$, obtained assuming zero surface buoyancy and non-zero interior PV anomaly ($b_s = 0$ and $q \neq 0$).

Assuming a constant stratification $N(z) = N_0$ and an ocean of depth $H$, the surface solution is (Tulloch and Smith, 2006)

$$\hat{\psi}_{srf}(k, z) = \frac{\hat{b}_s}{n_0 f_0 k} \frac{\cosh(n_0[H + z]k)}{\tanh(n_0 k H)}, \tag{39}$$

where ˆ stands for the Fourier transform, $\boldsymbol{k} = (k_x, k_y)$ is the wavevector, $k = \|\boldsymbol{k}\|$ its modulus and $n_0 \equiv f_0^{-1} N_0$, which becomes the classical Surface Quasi-Geostrophic solution in the limit $H \to \infty$ (Held et al., 1995; Lapeyre, 2017):

$$\hat{\psi}_{srf}(k, z) = \frac{\hat{b}_s}{n_0 f_0 k} \exp(n_0 k z). \tag{40}$$

The interior solution is

$$\hat{\psi}_{int}(k,z) = -\frac{\xi \hat{b}_s}{f_0 \left(k^2 + \frac{1}{n_0^2 H^2}\right)}, \tag{41}$$

which corresponds to the baroclinic mode (e.g. Klein et al., 2010). The relative dominance of each solution can be separated by a critical wavelength that depends on the large scale properties of the flow (Lapeyre, 2009; Klein et al., 2010). Additional expressions can be obtained taking, for example, an exponential stratification (e.g. LaCasce, 2012).

At the ocean surface, $\psi_{srf}$ dominates and projects onto $\psi_{int}$ (Lapeyre and Klein, 2006; LaCasce, 2012), which was used by Lapeyre and Klein (2006) to propose to approximate the total solution by a modified surface solution with an effective Brunt-Väisälä frequency $n_e$ that had to be adjusted using independent observations. Then, the three-dimensional geostrophic stream function and buoyancy can be retrieved from satellite measurements of SST as (Isern-Fontanet et al., 2008):

$$\hat{b}(\boldsymbol{k},z) = \frac{g\alpha_T}{\rho_0} \hat{T}_s \exp(n_0 k z) \tag{42}$$

$$\hat{\psi}_e(\boldsymbol{k},z) = \frac{g\alpha_T}{n_e \rho_0 f_0} \frac{\hat{T}_s(\boldsymbol{k})}{k} \exp(n_0 k z). \tag{43}$$

These equations are known as the effective SQG (eSQG) model. It is worth mentioning that, the parameter $n_e$ contains the contribution of interior PV as well as the effect of SSS, if salinity measurements are not used to derive the geostrophic velocities (see Isern-Fontanet et al., 2008). Moreover, using the relationship between SSH and the stream function (section 2.1), the above equations can be written for SSH (Isern-Fontanet et al., 2008)

$$\hat{b}_s(\boldsymbol{k},z) = n_e g k \hat{\eta} \exp(n_0 k z) \tag{44}$$

$$\hat{\psi}_e(\boldsymbol{k},z) = \frac{g}{f_0} \hat{\eta} \exp(n_0 k z) \tag{45}$$

Notice that, within this framework, SST and SSH contain the same information and, once buoyancy and the stream function are known at all depths, vertical velocities can be estimated (Lapeyre and Klein, 2006; LaCasce and Mahadevan, 2006; Klein et al., 2009; Isern-Fontanet and Hascoët, 2014). It has been shown that this approach can be used to derive ocean currents from real SST measurements (LaCasce and Mahadevan, 2006; Isern-Fontanet et al., 2006b) and SSS from SMOS (Isern-Fontanet et al., 2016b). Moreover, the eSQG approach has shown to provide good results in highly variable areas such as the Alboran Sea (Isern-Fontanet, 2016; Isern-Fontanet et al., 2017b) and for small ($\sim 10$ km) coastal eddies (Isern-Fontanet et al., 2017a). The validity of the eSQG approach has been extensively investigated using numerical models and real data (Lapeyre and Klein, 2006; Isern-Fontanet et al., 2006b, 2008, 2014; González-Haro and Isern-Fontanet, 2014; Qiu et al., 2016). Results show that the Mixed Layer (ML) depth is a good indicator of the periods in which the phase shift between SSH and SST is minimal, but different from zero, and, consequently, the eSQG approach can be applied (Isern-Fontanet et al., 2014). The best situations

correspond to deep ML, that are typically found in winter when smaller stratification favors the deepening of the ML (see Klein and Hua, 1990, for a discussion on the effect of ML deepening on SST). Notice that this approximation has a limited capability to reconstruct the vertical structure of the ocean (e.g. Isern-Fontanet et al., 2008; LaCasce, 2012) which has lead to propose improved models of the upper ocean dynamics (Wang et al., 2013; Ponte and Klein, 2013; Chavanne and Klein, 2016). These models, however, require the knowledge of the geostrophic stream function at the ocean surface, which is the sought field here.

The comparison between altimetric measurements of SSH and SST images unveils the synergy between these two measurements (e.g. figure 4) . In general, while SST images can be used to obtain information about the location and geometry of ocean structures, it is difficult to quantify velocities from them (see also section 2.3). Conversely, although altimeters provide information about ocean velocities, it is difficult to recover the location and geometry of ocean structures. However, within the eSQG framework, SSH and SST are in phase and contain the same information. These ideas motivated Isern-Fontanet et al. (2014) to reconstruct the surface stream function combining SST and SSH measurements through the definition of an empirical transfer function, $F(k)$:

$$\hat{\psi}_s(\boldsymbol{k}) = F(k)\hat{T}_s, \tag{46}$$

where $F(k)$ can be empirically estimated combining SST and SSH measurements as

$$F(k) \approx \frac{g}{f_0} \frac{\langle|\hat{\eta}|\rangle_k}{\langle|\hat{T}_s|\rangle_k}. \tag{47}$$

This idea has been analyzed in Isern-Fontanet et al. (2014) and González-Haro and Isern-Fontanet (2014) that showed that the transfer function can be approximated by a Butterworth filter

$$F_b(k) \approx A \left[1 + \left(\frac{k}{k_c}\right)^{2\gamma}\right]^{-\frac{1}{2}} \tag{48}$$

with $\gamma = 1$, $k_c$ a cut-off frequency and $A$ an amplitude that has to be determined from other measurements such as altimetric data, drifters, etc (equivalently to the $n_e$ parameter in the eSQG approach). This approach is well suited to combine simultaneous measurements of SST and SSH such as the ones provided by Sentinel-3 satellite from ESA.

During the recent years there have been some efforts to include the ageostrophic effects into the SQG framework. On one side, Ponte et al. (2013) included wind-driven ageostrophic contributions into the SQG dynamics. They integrated equation 7 (without the buoyancy and Stokes terms) over a ML of depth $h$, using pressure derived from SSH and assuming and SQG-like vertical decaying (equation 45) and the parameterization of the turbulent stress given by equation 10:

$$\hat{\boldsymbol{v}}(\boldsymbol{k}) = \frac{\hat{\boldsymbol{v}}_0}{n_e f_0 k h} \left[1 - \exp(-n_0 k h)\right] \tag{49}$$

where $\boldsymbol{v}_0(\boldsymbol{x})$ is the geostrophic velocity at the surface. Interestingly, the effect of wind does not appears explicitly in the above equation and is contained in the ML depth. Moreover, this solution implies that at scales smaller than those of wind stress, i.e a few hundreds of kilometers, the total averaged velocity is in phase with the geostrophic velocity. On the other side, Badin (2013) also included ageostrophic effects by re-writing the SQG using the two-dimensional semi-geostrophic equations allowing to extend the this approach to scales smaller than the Rossby radius of deformation.

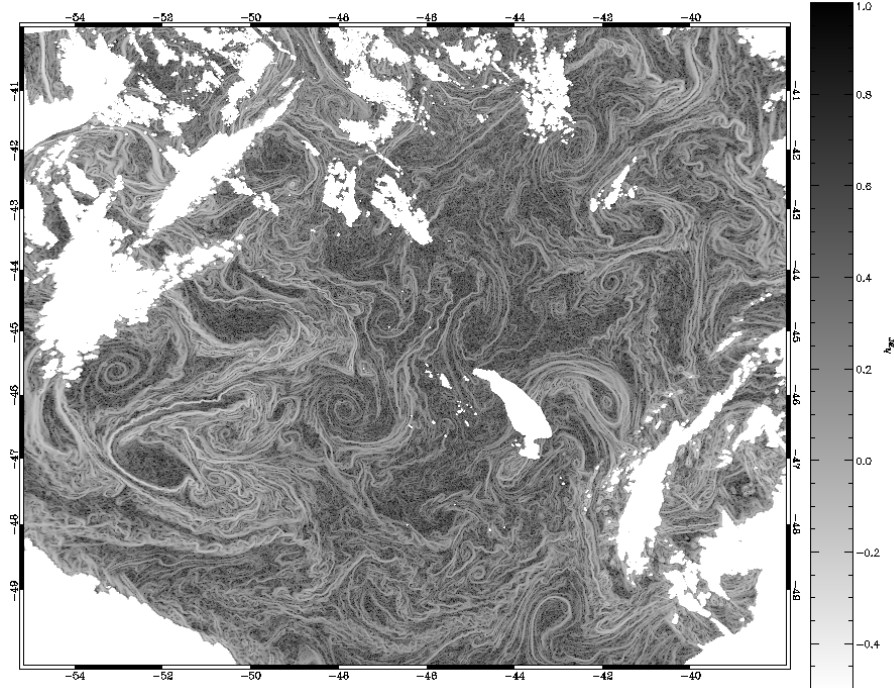

**Figure 7.** Singularity exponents derived from the Brightness Temperature of the image shown in figure 4.

Besides the use of PV inversion arguments, the identification between frontal structures and stream-lines has also been exploited to derive ocean currents from a single SST image. In particular, it has been explored the use of Singularity Analysis (Turiel et al., 2005; Isern-Fontanet et al., 2007; Turiel et al., 2008). Singularity exponents are dimensionless variables that measure the local degree of regularity (if positive) or irregularity (if negative) of the scalar at each point. The set of singularity exponents do not only provide information about the statistics of changes of scale in the scalar, but also about the specific geometrical arrangement of the structures explaining those changes in scale. A striking feature of singularity exponents is that singularity isolines, especially those associated to the most singular values (i.e., more negative), seem to delineate with remarkable accuracy the streamlines of the flow. They do so more closely than the isolines of the scalar from which they are derived (see, for instance, figure 8 in Turiel et al. (2009)). However, no theoretical proof of this observed property has been given so far. Figure 7 shows for the matter of example the map of singularity exponents derived from the SST map shown in 4. As shown in the figure, the singularity exponents provide very detailed information about the patterns underlying the SST, and provide a constant, homogeneous value along singularity lines, despite the progressive change in the amplitude of the gradient of SST. Fronts and sharp transitions in general are associated with negative values and so they are shown in white colors in the figure, but also subtler transitions (i.e., smaller amplitude gradients) are associated to negative values, what allows to uncover a more detailed view of the circulation. Positive values (represented in dark colors in the figure) are also in correspondence with frontal structures but which have less dynamic relevance.

The apparent correspondence between singularity lines and streamlines motivated the introduction of a simple method (called Maximum Singular Stream function Method or MSSM, Turiel et al. (2005); Isern-Fontanet et al. (2007)) that provides an estimate of a normalized stream function from the singularity exponents obtained from a map of a given ocean scalar. However, the MSSM is not very useful for dynamic studies, as it just gives information on the geometry of the flow, but neither the modulus of the velocity vector nor the sense of the circulation (upstream or downstream the depicted streamlines) are known. Besides, by construction the MSSM relies in the capability of the so-called Most Singular Manifold (MSM) to describe the full geometry of the flow, something that introduces a certain degree of quality loss in the method due to numerical degradation. Nevertheless, the capability of singularity analysis to capture the underlying organization of the flow points to its future combination with the SQG approach or with altimetric data to improve the reconstruction of high-resolution velocities.

## 3  Retrieval from coastal HF Radars

The lack of direct satellite measurements of surface ocean currents has motivated the development of different techniques to derive them from complementary satellite observations as seen in section 2. These techniques are based on imposing theoretical frameworks that are a simplification of the dynamics, even to respect the dynamics underlying current ocean models. An alternative to avoid this issue is to use coastal radars, which allow remote sensing retrievals of ocean currents by measuring the Doppler shift of the radio waves back-scattered by small sea surface waves. Radars operating in the 3-50 MHz range have the advantage that the emitted wavelengths (6 m to 100 m) are comparable to those of typical surface waves, translating to a strong backscatter (Paduan and Rosenfeld, 1996). As the frequency range includes the High Frequency (HF) band of the electromagnetic spectrum, these radars are called HF.

Two methodologies are presently being used: the CODAR SeaSonde (Barrick, 2008) and the Wellen radar (WERA, Gürgel et al., 1999), being the differences between them the configuration for retrieving both the speed and direction. HF radar systems in coastal areas have rapidly evolved during the first decade of this century and presently the global network is composed of roughly 170 sites mostly in the west and east coast of the US and with lesser extent in Europe (Rubio et al., 2017) and Australia (figure 8).

Radar-derived currents are assumed to have a measurement depth of 1 m at 10-15 Mhz, and they have been extensively used for oceanographic studies in coastal regions. See the exhaustive review by Paduan and Washburn (2013) and the references therein.

HF radars provide spatial and temporally averaged currents. They retrieve their information from a horizontal footprint that changes with the distance from the antenna. Although they can provide information of the surface currents up to 20-70 km from the coast, the actual coverage depends on radio interferences, the time of the day, solar activity, and sea state (Paduan and Washburn, 2013). The frequency spectra of any radar measurement reveals the existence of white noise (Forget, 2015). The amplitude of the noise is not linked to the radar station, as it changes with time and location. In its analysis, Forget (2015) concludes that the average sampling period should have to adapt in order to retrieve the geophysical signal. The origin of such noise has not yet been fully understood, but various processes have been proposed to affect the radar measurements:

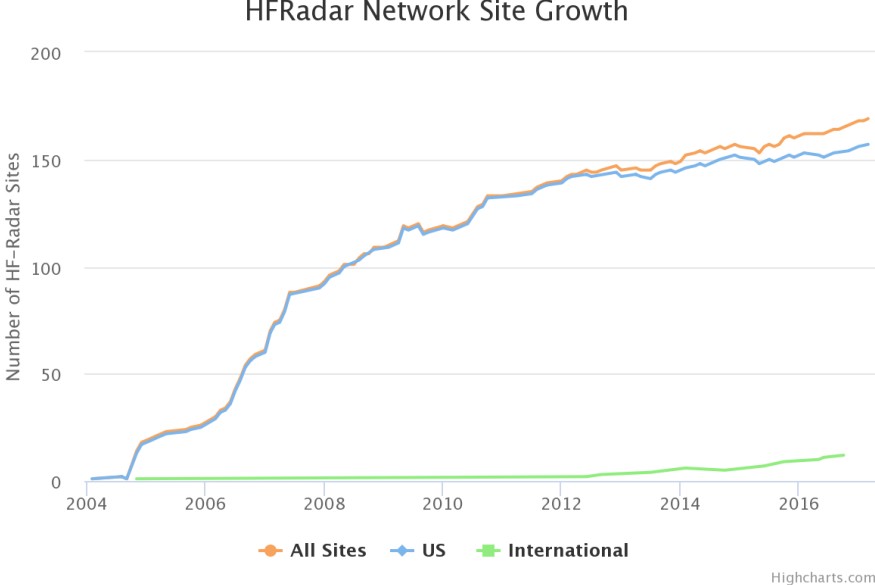

**Figure 8.** Growth of HF radar sites. Source: Coastal Observing Research and Development Center (CORDC), available at http://cordc.ucsd.edu/.

changes in the velocity field during the duration of the radar measurement; (Lipa et al., 2006), Radio frequency interferences ịtepMerz.2015; antenna pattern (Lipa et al., 2006); and signal sampling (Liu et al., 2014).

The effective spatial resolution of long-range radar systems has been investigated by Heron and Atwater (2013). Their analysis indicates that the effective resolution of WERA antennas ranges from the 10 km near to the radar stations and 25 km
at long range (150 km). The resolution of SeaSonde antennas is 40 km and 60 km respectively.

Being an integrated measurement, the nature of the radar-derived currents remains an open debate. For example, it has been suggested that HF radar currents include the entire wave-induced Stokes drift (Graber et al., 1997), part of it (Ardhuin et al., 2009) or none of it (Röhrs and Christensen, 2015). In their work, Röhrs and Christensen (2015) compare HF radar currents with two types of surface drifters: seven iSphere drifters without drogue (found to be driven by the Eulerian current
and the Stokes drift at surface) and seven CODE-type drifters (following the ocean current at 1 m depth). Both types of drifters experimented little wind drag. In their comparison they found that the difference between HF radar currents and the iSphere velocities strongly correlated with the Stokes drift. Moreover, the difference between HF radar velocities and the CODE-type drifters appeared to be independent of Stokes drift for the wind and wave conditions in their study area.

The results of Röhrs and Christensen (2015) indicate that the drifters responding to the vertically integrated surface currents
might be more suitable for HF radar validation than drifter without drogue, although they caution that the results might depend on the local dynamics.

## 4 Data assimilation of ocean currents

In this section we will focus on the various applications assimilating remote sensed ocean velocities in regional and coastal simulations. In most of the following applications, ocean currents are mainly derived from coastal HF radars, and only two works refer to the assimilation of global currents derived from altimeter data.

In the case of coastal simulations, it is widely accepted that the main source of errors is the inadequate wind stress forcing. Assimilation of HF radar could improve the realism of the simulations by partially correcting surface wind forcing. However, the amount of available observations (HF radar, along-track altimetry and SST maps from satellites and vertical temperature and salinity profiles from moorings, gliders and profilers) remains sparse compared with the fast, small-scale, nonlinear dynamics characteristic of coastal areas.

The first work assimilating HF radar surface data into an ocean model was done by Lewis et al. (1998) using a nudging technique to correct the model surface current towards the HF radar estimates. Since then, and driven by the continuous expansion of the network of HF radar systems, different data assimilation approaches have been used to assimilate HF radar currents into non-linear, high-resolution ocean models: nudging (Lewis et al., 1998; Wilkin et al., 2005; Gopalakrishnan and Blumberg, 2012), sequential assimilation (Breivik and Sætra, 2001; Oke et al., 2002; Paduan and Shulman, 2004; Kurapov et al., 2005a; Oke et al., 2009) and four-dimensional variational (4DVAR) assimilation schemes (Hoteit et al., 2009; Zhang et al., 2010; Yu et al., 2012).

### 4.1 Nudging

The first work aiming to assimilate HF radar currents into a regional model of the Monterey Bay (California, US) was published by Lewis et al. (1998). The HF radar observations, $\boldsymbol{u}^o$, were assimilated by adding a fictitious surface wind stress term that nudged the model solution $\boldsymbol{u}_1$ (uppermost layer) towards the observed values:

$$\boldsymbol{\tau}(t) = \rho \, C_D \left( \boldsymbol{u}^o(t) - \boldsymbol{u}_1(t) \right) \left| \boldsymbol{u}^o(t) - \boldsymbol{u}_1(t) \right|, \qquad \forall t \tag{50}$$

with $\rho$ being the water density and $C_D$ a drag coefficient. The data being assimilated was the 30-minute averaged surface currents, available every two hours and linearly interpolated to the time step of the model. They showed that such a continuous assimilation strategy was able to modify the model currents towards the observed direction. However, significant differences remained in the velocity field even after more than 170 hours of assimilation. In particular, the reconstructed velocities remained small compared with the observed ones. The authors pointed out that errors in the Doppler retrieved currents might have been the reason for it and suggested that the HF data should be processed before assimilation. For example, by removing the divergent component from the observation field. The same approach was used by Santoki et al. (2013) to assimilate $1° \times 1°$ OSCAR currents (see section 2.2) in a basin-wide simulation of the Indian Ocean. In this work, the current measurements from three RAMA buoys were used to assess the impact of the assimilation. The authors pointed out that, although it is said that OSCAR currents do not provide an accurate representation of the meridional currents at these RAMA locations, the model performed even worse. The assimilation of OSCAR velocities reduced the deficiencies of the model at these locations (figure 9).

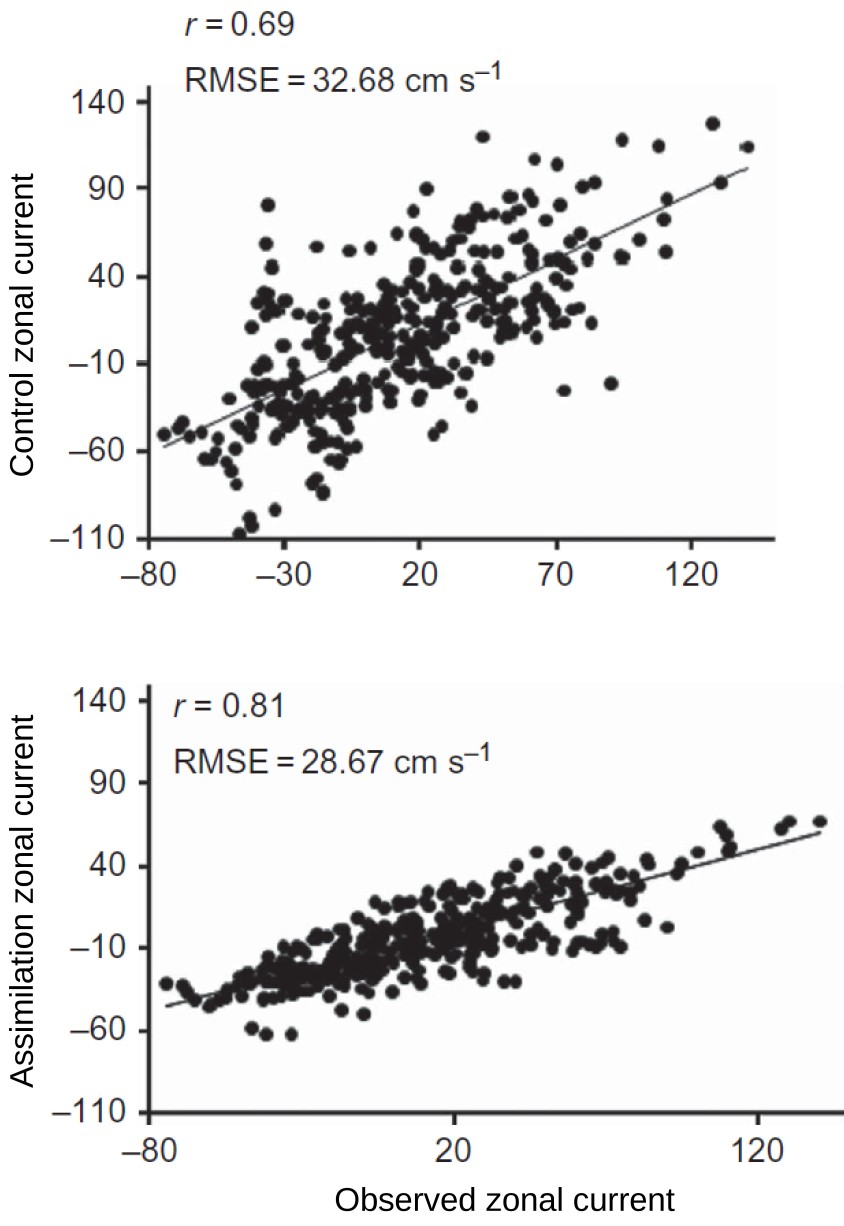

**Figure 9.** Correspondence between the zonal velocity component measured at the RAMA station located at 1.5° N 90.0°E. Upper plot model without assimilation. Lower plot, resulting from assimilating OSCAR currents.From figure 1 in Santoki et al. (2013).

A strategy to simultaneously update the 3D velocity field was used by Wilkin et al. (2005) in the New Jersey coast (US). In their application, they estimated the correlation between the surface CODAR data and the measurements provided by a moored Acoustic Doppler Current Profile (ADCP) and used them to project surface CODAR data to the depth. The authors

compared two methodologies to feed their 3D maps into the dynamical model: a continuous nudging and the intermittent melding described by Dombrowsky and De Mey (1992). Their results indicate that the intermittent corrections of the 3D ocean currents better allowed the model to freely adjust and develop than the continuous nudging of the model observations toward observations.

The nudging scheme of Gopalakrishnan and Blumberg (2012) used a four-dimensional nudging coefficient:

$$\frac{\partial \boldsymbol{u}(\boldsymbol{r},t)}{\partial t} = \{Physics\} +$$
$$\sum_{i=1}^{p} \mu(\boldsymbol{r}_i^o - \boldsymbol{r}, t_i^o - t) \left[ \boldsymbol{u}_i^o(\boldsymbol{r}_i^o, t_i^o) - \boldsymbol{u}(\boldsymbol{r},t) \right], \tag{51}$$

where the nudging coefficient, $\mu$, was a function of the distance between the observations and each model grid point. In their work, they propose an analytic form for the nudging coefficient:

$$\mu(\boldsymbol{r}^o - \boldsymbol{r}, t^o - t) = \mu_o \, e^{-\left(\frac{\Delta r_H}{R_H}\right)^2} e^{-\frac{|z|}{Z_d}} e^{-\frac{|\Delta t|}{T_d}}, \tag{52}$$

where $\Delta r_H$ is the horizontal separation between $\boldsymbol{r}^o$ and $\boldsymbol{r}$, $R_H$ is the nudging length-scale, $Z_d$ is the depth of influence of the surface observation and $T_d$ is a damping time-scale. Each observation may accelerate and decelerate a fraction of the water column, disseminating the corresponding stresses in the four-dimensional neighborhood of each observation. In their application to assimilate HF radar data in the Raritan Bay and the coastal waters of New York and New Jersey, they

implemented the limiting case $R_H \to 0$, $T_d \to 0$, $\mu_o = (1800\,s)^{-1}$ and $Z_d = 2\,m$. The impact of the assimilation was estimated using in situ observations of the ocean currents, temperature and salinity withheld from the assimilation. They found that the vertically-projected nudging was able to improve both the hindcasting and the 24-hour forecasts of near-surface currents and temperature.

## 4.2  Sequential methods

Breivik and Sætra (2001) used what they called a "quasi-ensemble" assimilation scheme derived from the Ensemble Kalman Filter (EnKF) introduced by Evensen (1994) to assimilate HF radar observations into a 1-km, nested, regional model of the Fedje area (Norway). The basic equations of the EnKF are:

$$\boldsymbol{x}^a = \boldsymbol{x}^f + \boldsymbol{K} \left[ \boldsymbol{y}^o - H\boldsymbol{x}^f \right], \tag{53}$$
$$\boldsymbol{K} = \boldsymbol{P}^f H^\top \left[ H\boldsymbol{P}^f H^\top + \boldsymbol{R} \right]^{-1}, \tag{54}$$
$$\boldsymbol{P}^f = \frac{\alpha}{r-1} \boldsymbol{X}' \boldsymbol{X}'^\top. \tag{55}$$

In equation 53, $\boldsymbol{x} \in \mathbb{R}^n$ represents the $n$-dimensional model *state vector*. In an ocean model, the state vector is usually constructed from the values of sea level, and the three-dimensional fields of temperature, salinity, horizontal currents. The superscripts $a$ and $f$ indicate the analysis and the forecast solutions respectively. The vector $\boldsymbol{y}^o \in \mathbb{R}^p$ represents the set of $p$ observations available at the analysis time. The *observation operator*, $H : \mathbb{R}^n \to \mathbb{R}^p$, projects the model solution to the ob-

servation space. When the observation operator is linear, it is represented by the observation matrix $\boldsymbol{H} \in \mathbb{R}^{n \times p}$. The model

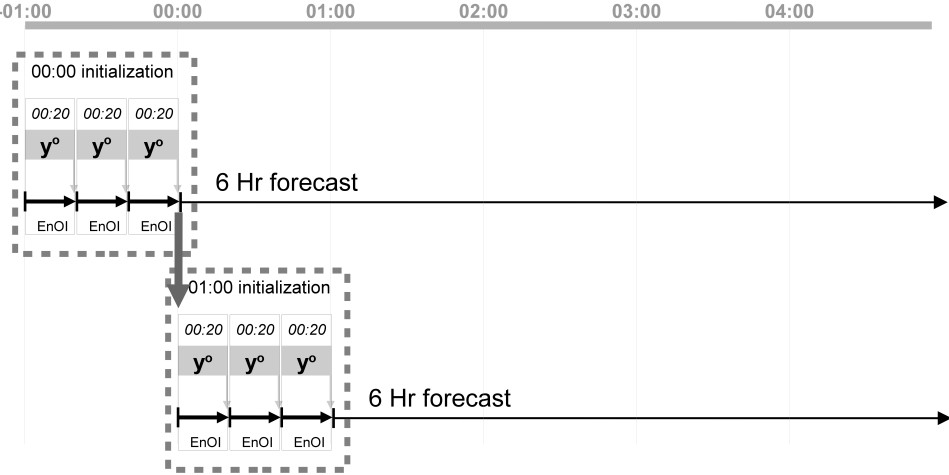

**Figure 10.** Data assimilation cycle in Breivik and Sætra (2001). Surface currents are used to initialize, every hour, a 6-hour prediction. In the initialization procedure, three cycles of EnOI are used to assimilate the current data available every 20 minutes.

error covariance matrix is given by $\boldsymbol{P}^f \in \mathbb{R}^{n \times n}$. Similarly, the observation error covariance is given by $\boldsymbol{R} \in \mathbb{R}^{p \times p}$. The matrix $\boldsymbol{K} \in \mathbb{R}^{n \times p}$, called the *Gain matrix*, extrapolates the information from the observation locations to every component of the state vector. As such, equation 53 has the potential to correct the state of the whole three-dimensional system from a set of observations of the surface current. The term $\boldsymbol{K}[\boldsymbol{y}^o - H\boldsymbol{x}^f]$ is known as the *assimilation increment* and it is used to project, to

the model space, the information provided by the observations that was missing in the forecast.

     The gain matrix $\boldsymbol{K}$ given by equation 54 is said to be *optimal* (in the sense that it provides the most likely estimate of the system provided the values being observed) if the system is linear and if both forecast and observation errors are Gaussian and unbiased. However, as discussed by Evensen (1994), this is not the case when the system dynamical laws are non-linear. Indeed, in non-linear systems, the time evolution of Gaussian errors is not longer Gaussian, and the error covariance matrix does no

longer fully describe the statistical properties of the forecast errors. For non-linear models, Evensen (1994) proposes equation 55 as a Monte-Carlo estimation of the forecast error from the dispersion of an ensemble of plausible estimates of the state of the system. Specifically, let us consider an ensemble of $r$ model states, $\boldsymbol{x}_i(t), i = 1, \ldots, r$, evolving according to the non-linear system dynamics and differing because of differences in the initial conditions, external forcing or model parameters. At any time, $t$, the ensemble mean, $\overline{\boldsymbol{x}}(t) = (1/r)\sum_{i=1}^{r} \boldsymbol{x}_i(t)$, and the ensemble of anomalies, $\boldsymbol{x}'_i(t) = \boldsymbol{x}_i(t) - \overline{\boldsymbol{x}}(t)$, can be easily

calculated. If we define the matrix $\boldsymbol{X}'(t) \in \mathbb{R}^{n \times r}$ as the matrix whose columns correspond to the members of the ensemble of anomalies,

$$\boldsymbol{X}'(t) = [\boldsymbol{x}_1(t) - \overline{\boldsymbol{x}}, \boldsymbol{x}_2(t) - \overline{\boldsymbol{x}}, \cdots, \boldsymbol{x}_r(t) - \overline{\boldsymbol{x}}], \tag{56}$$

the ensemble covariance is given by equation 55.

An advantage of the EnKF is that, at each time step, we can easily calculate the projection of the state vector onto the observation space:

$$H\boldsymbol{X}'(t) = [H\boldsymbol{x}_1(t) - \overline{H\boldsymbol{x}}, H\boldsymbol{x}_2(t) - \overline{H\boldsymbol{x}}, \cdots, H\boldsymbol{x}_r(t) - \overline{H\boldsymbol{x}}], \tag{57}$$

a fact that allows the calculation of the terms $H\boldsymbol{P}^f H^\top$ and $\boldsymbol{P}^f H^\top$ without the need of explicitly estimating the error covari-
ance matrix $\boldsymbol{P}^f$ (equation 55) or the operator $H^\top$ (Houtekamer and Zhang, 2016). This fact strongly reduces the computational cost associated with 54.

The parameter $\alpha$ in equation 55, known as *inflation factor*, is introduced to scale the weight of the ensemble versus the observations, to take into account the effect of the model error, and to avoid the collapse of the covariance matrix. To reduce the impact of the sampling errors (i.e., the errors arising from the fact of using a finite ensemble) in the estimation of ensemble
covariance matrices, some kind of localization is usually used to reduce the effect of spurious covariances. An example example of the pervasive effects of the spurious covariances in systems with short and long scales can be found in Ballabrera-Poy et al. (2009). Covariance localization can be explicitly implemented by multiplying the empirical covariance by an analytic localization function (Hamill et al., 2001) or by performing a local analysis in which we divide the state space into a set of independent local analysis domains, limiting the influence of observations to some subset of space points or state variables
(Cohn et al., 1998). Implicit implementation of localization is obtained by truncating the eigenvalue expansion of the term $H\boldsymbol{P}^f H^\top + \boldsymbol{R}$ in equation 54 (Oke et al., 2002).

The quasi-ensemble proposed by Breivik and Sætra (2001) consisted of replacing the ensemble of model simulations with an ensemble of model states coming from a unique model simulation taken at different times:

$$\boldsymbol{X}' = [\boldsymbol{x}'(t_1), \boldsymbol{x}'(t_2), \cdots, \boldsymbol{x}'(t_r)]. \tag{58}$$

A necessary condition for the ensemble 58 to have a meaningful covariance 55 is that the collection of states defining the *ensemble* is taken from a representative model simulation. The advantage of using equation 58 is that, once the ensemble has been constructed, the covariance remains constant, reducing the numerical cost of the assimilation algorithm 53-55. The resulting algorithm has been known lately as an Ensemble Optimal Interpolation (EnOI, Evensen, 2003).

In Breivik and Sætra (2001), the radar data was available every 20 minutes, and three data assimilation cycles were used to
get the initial conditions for a 6 hour forecast (Figure 10). The low cost of the EnOI made possible to have a 6-hour forecast within 45 minutes since the data acquisition time. However, although equation 53 allows the correction of the three-dimensional hydrographical fields of the model (temperature and salinity), Breivik and Sætra (2001) found that the model rapidly became unstable. The reason was the nested nature of the simulation. Without correcting the external, coarse simulation, large density gradients built up between the (free) external and the (constrained) internal simulations. Therefore, they had to leave out the
cross-updates of temperature and salinity. As such, the information added by the assimilation was lost after 6 hours. Years later, Zhao et al. (2013) compared the approach of Breivik and Sætra (2001) with the usual implementation of the EnKF (Evensen, 1994), in an experiment assimilating hourly surface currents over the Qingdao coastal waters (China). In Zhao et al. (2013), the ensemble members corresponded to the difference between successive model outputs every 6 hours during one month. Their

results indicated that, although EnKF provides a better fit to independent surface currents, both EnOI and EnKF improve the simulation of the coastal surface currents.

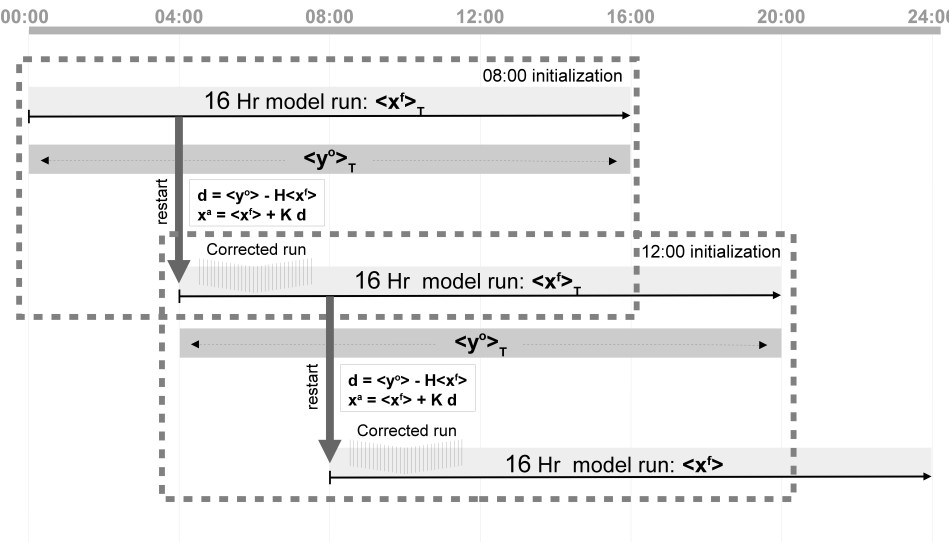

**Figure 11.** Data assimilation cycle in Oke et al. (2002). The time-distributed averaging procedure approach used to initialize the problem at time T/2 uses all the observations in the period [0,T]. In their application, the time T is approximately the inertial period.

Another seminal implementation of the EnKF to assimilate a subset of observations from an array of CODAR SeaSonde HF radars deployed along the Oregon coast was described by Oke et al. (2002). In their work, they used a stationary version

5 of the Physical-space Statistical Analysis System (PSAS) introduced by Cohn et al. (1998) and a Time-Distributed Averaging Procedure (TDAP). Observations were low-pass filtered to remove the tidal signal, and the average during a full inertial period $[0,T]$, i.e. approximately 17 hours, was assimilated using an EnOI algorithm to obtain an estimate of the system at time T/2 (Figure 11). The model was then initiated at time T/4 from a true solution of the model and ran until T/2. At each time step, the model solution is corrected as

$$ \boldsymbol{x}(k\Delta t) = \boldsymbol{x}(k\Delta t) + \frac{1}{N_k}\boldsymbol{K}\left(<\boldsymbol{y}^o>_T - H<\boldsymbol{x^f}>_T\right), \tag{59}$$

where $k = 1, \ldots, N_k$ refers to the time steps of the simulation. One of the advantages of the time distributed strategy is that the model always starts from a pure model output, avoiding initialization shocks. As the assimilation increment is distributed over a quarter of the inertial period, it allows the model dynamics to adjust to the data assimilation increment, better preserving the model dynamical balances. The results were validated using data from a moored Acoustic Doppler Current Profiler (ADCP).

15 The authors found that, despite the presence of an unexplained bias in the results, the data assimilation increased the magnitude of the fluctuations of the model velocity field increasing the agreement with the observations (figure 12). The authors pointed out that the assimilation of HF radar data compensated for the unrepresented signal of the wind stress forcing used in their simulation.

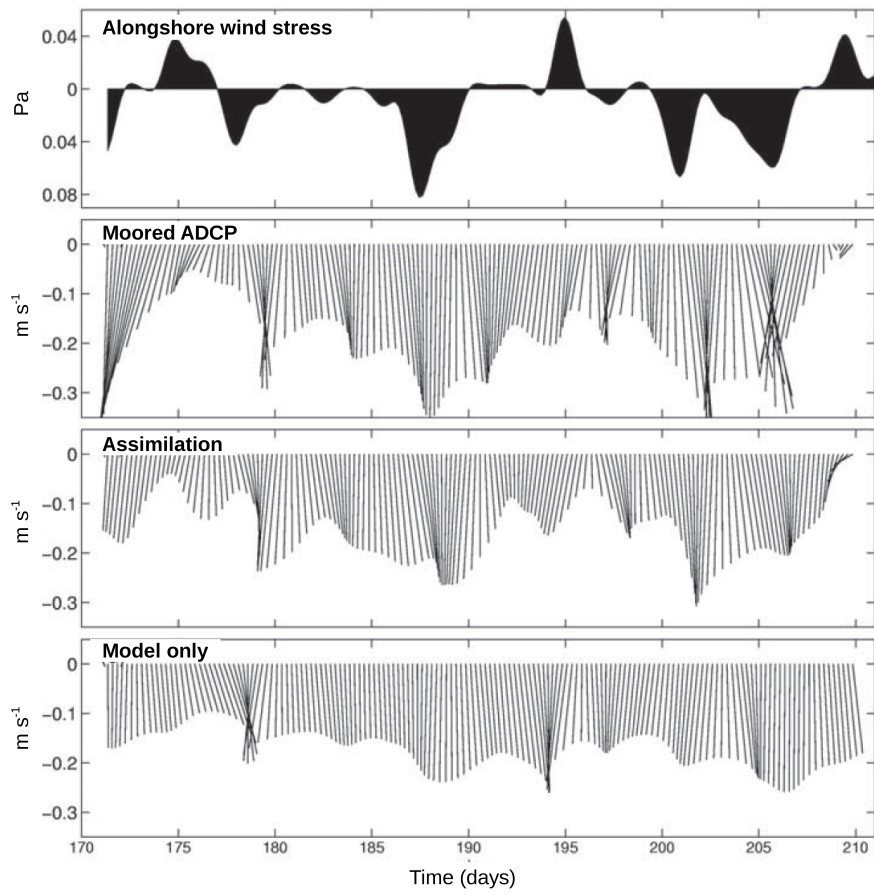

**Figure 12.** Comparison between the alongshore wind stress and the ocean vertical averaged current during the 40-day experiment. From figure 10 in Oke et al. (2002).

Paduan and Shulman (2004) assimilated low-pass filtered Monterey Bay HF radar measurements using a two-step data assimilation approach: they used an EnOI method to update the velocity field of the first layer of the model, and a second step in which the surface velocity corrections were projected downward using Ekman theory arguments of either energy conservation or momentum transfer. They illustrated the disadvantage of only correcting the surface layer as had been done in Lewis et al. (1998). The simultaneous correction of the 3D velocity field reduced the spurious velocity shear that occurs when only the surface layer of the model is corrected.

Kurapov et al. (2005a, b) used an approach similar to Oke et al. (2002) to assimilate velocity profiles measured by a set of moorings in a regional simulation of the Oregon coast. As in Wilkin et al. (2005), only the velocity field was updated and the other variables were allowed to evolve as a result of the dynamical adjustment. Disregarding the ensemble covariance between currents and the hydrography fields was justified by the weak correlation that existed between these variables but also because

of the sampling error of the empirical correlations estimated by the EnOI. Their results showed that their EnOI algorithm was able to improve the solution of the model and to induce significant dynamical changes.

A slightly different approach was used by Barth et al. (2008) to assimilate 2-day averaged currents in a nested simulation of the West Florida Shelf. Only the radial HF radar component was seen by the data assimilation algorithm, and the back-
ground error covariance is used to statistically extrapolate the velocity perpendicular to the radial direction. In their work the background error covariance matrix was built from a set of model simulations differing in the wind forcing. The reference wind forcing combines the NCEP NAM (North American Mesoscale Model) with *in situ* wind measurements. The 6-hour wind field during the year 2004 was used to calculate a set of Empirical Orthogonal Functions (EOFs). An ensemble of 100 synthetic wind fields was created by perturbing the reference wind field with a linear combination of these EOFs with Gaus-
sian random coefficients. The analysis step corrected both currents and hydrography. Similar to the findings of Lewis et al. (1998), the authors found that the forecast skill improved if a spatial filter is used to remove spurious barotropic waves from the assimilation increment and if the wind stress is included in the state vector. This allows the data assimilation to correct both the state of the ocean and the forcing term. In Barth et al. (2011), a similar ensemble approach is implemented with a state vector that contained only the wind forcing of the model, i.e. $\boldsymbol{x} = (\boldsymbol{\tau}_x, \boldsymbol{\tau}_y)$. In that case, the implicit observation operator
provides the corresponding upper ocean surface current, i.e. $H\boldsymbol{x} = \boldsymbol{u}_1$. The rationale behind this approach was the thought that too frequent assimilation of observations often produces unrealistic features that, if not dissipated, will degrade the model results. They opted for correcting the main source of the model error (the wind stress forcing) rather than the state of the ocean itself. Their results were validated against independent wind and SST observations. Their results indicate that improvements in the amplitude of the wind stress drove the corresponding improvement in the SST. However, in places where the SST was
driven by other factors (e.g., open boundary conditions), changes in the forcing wind had no impact. The effort of using HF radar measurements to correct (separately) wind forcing and the open boundary conditions was done by Marmain et al. (2014). In both cases, although some reduction of the error was obtained for surface currents, mixed results were obtained by respect temperature and salinity.

The expected advantage of incorporating HF radar and *in situ* temperature and salinity observations from glider transects
into the operational system used by the Australian Bureau of Meteorology was investigated by Oke et al. (2009). They used the Bluelink Ocean Data Assimilation System (BODAS), an EnOI data assimilation system descendant from the pioneering work of Oke et al. (2002). Using synthetic HF radar and gliders, they checked the added value that these observations would have in their operational system. They found that HF data could reduce the analysis errors by 80%, with improvements reaching 200 km beyond the radar footprint. Moreover, as HF radars are able to detect spatial structures smaller than the ones resolved
by the Global Ocean Observing System, they would also help reduce sea level errors. However, glider transects were found to have only a localized impact, probably due to the short spatial scales over the shelf region. It was thus suggested that, if a glider program was to be implemented, transects should be closely spaced (around 100 km) to resolve the mesoscale variability.

## 4.3 4DVAR

Hoteit et al. (2009) used a four-dimensional variational (4DVAR) approach using the Massachusetts Institute of Technology general circulation model (MITgcm) introduced by Marshall et al. (1997) to dynamically interpolate HF radar data collected off the San Diego coast. Application of 4DVAR algorithms always start by defining a cost function of the type:

$$
J(\boldsymbol{u}) = \sum_{t=0}^{T} \left[\boldsymbol{y}^o(t) - H\boldsymbol{x}(t)\right]^\top \boldsymbol{R}^{-1} \left[\boldsymbol{y}^o(t) - H\boldsymbol{x}(t)\right] +
$$
$$
\sum_{t=0}^{T} \left[\boldsymbol{u}(t) - \boldsymbol{u}^b(t)\right]^\top \boldsymbol{Q}^{-1} \left[\boldsymbol{u}(t) - \boldsymbol{u}^b(t)\right],
$$
(60)

which is a weighted average of the model-data misfit and the changes to the control variables. The *control vector* $\boldsymbol{u}(t)$ must be defined according to each particular application. It usually contains the initial model state (currents, temperature and salinity), the fields at the open boundaries,atmospheric forcing fields (mass and momentum) or model parameters. Note that if the initial model state is the only control variable, then error covariance matrix $\boldsymbol{Q}$ should be equal to the model error covariance $\boldsymbol{P}^f$ used in the EnKF. As such, the first term in equation 60 is a measure of the distance between the mode and the observations and the second term introduces penalties upon departures from the set of background control values $\boldsymbol{u}^b$. The goal of the 4DVAR is to find the optimal value of the control, $\boldsymbol{u}^*$, for which the cost function 60 reaches its mimimum value. For linear and perfect systems, it has been shown that the solution that minimizes equation 60 can be written as 53-54. See Lorenc (1986) for a detailed discussion. In the 4DVAR assimilation, the cost function is minimized iteratively. At each iteration, the ocean model is run forward to calculate the value of the cost function and its *adjoint model* is run backwards to obtain the gradient of the cost function by respect the control vector, $\nabla_{\boldsymbol{u}} J$, which is used to determine a descent direction towards the minimum (LeDimet and Talagrand, 1986).

Although not explicitly noted, the observation operator $H$, the observation error covariance, $\boldsymbol{R}$ and the error covariance matrix, $\boldsymbol{Q}$ should be a function of time, although in many applications (i. e. operational implementations) these matrices are kept constant. The specification of the error covariance matrix, $\boldsymbol{Q}$, is key in the performance of the 4DVAR system as it it introduces constraints in the space of all possible control values. In They usually are non-diagonal matrices to include geophysically balanced control values. Finding their appropriate form remain a research issue. Because of the lack of an appropriate observing system, physical, statistical and computational constrains usually dictate their form (Weaver et al., 2005). In particular, when control variables contain physical fields (e.g. the initial conditions), the covariance matrices are modeled using recursive filters (Lorenc, 1992), diffusion equations (Weaver and Courtier, 2001) and simplified linear balance operators (Dobricic and Pinardi, 2008).

In Hoteit et al. (2009) the model starts from rest and it is initialized using data from a single profile of T and S. The model is initially forced with wind data from a single shore station and with zero heat and fresh water fluxes. The model covers the San Diego coast region (US), has open boundaries in the north, west and south, and it does not include tides. The hourly HF radar velocities were then used to try to constrain the initial conditions, the open boundary conditions and the air-sea fluxes of heat, mass and momentum. The tidal components of the currents were removed using a least-square fit to four diurnal and

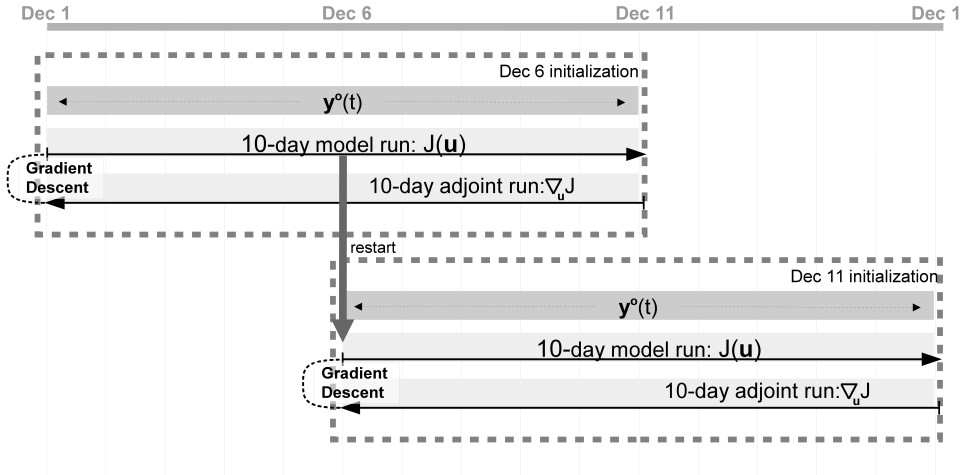

**Figure 13.** Data assimilation cycle in Hoteit et al. (2009). The pair of direct model run and adjoint model run is repeated iteratively until the pre-defined convergence criteria is reached. After convergence, the solution at the center of the assimilation period is used as the restart point for the next assimilation cycle: Overlap of five days.

four semi-diurnal tidal lines over a 1-year period. Their results showed that the observed surface currents could be fitted by adjusting the wind stress controls and that the resulting surface currents showed skill over persistence for about 20 hours. However, they found that without constraining the surface winds, the resulting solution was weakly sensitive to the control of initial and boundary conditions after about two inertial periods. Moreover, and similarly to the findings of previous works
using different data assimilation methods, they concluded that surface current observations alone were not enough to constrain the three-dimensional structure of the system.

The first implementation of a multivariate assimilation of multiple data sources including HF radar currents was done by Zhang et al. (2010) in the New York Bight using the Regional Ocean Modeling System (ROMS) model (Haidvogel et al., 2008) and its adjoint model (Di Lorenzo et al., 2007). Their data assimilation method was an incremental strong-constrain
4DVAR (Powell et al., 2008) that only adjusted the initial conditions using assimilation windows of three days, overlapping the data assimilation windows, advancing the beginning of the data assimilation window by one day. Using a series of sensitivity experiments they revealed that the assimilation of HF radar currents in the model increased the current prediction skill of the model by 1-2 days. However, assimilation of surface currents slightly degraded the prediction skill of subsurface temperature. These results indicated either the presence of deficiencies in the error covariance matrix, $Q$, used by the assimilation algorithm
or deficiencies in the dynamical model itself (and its forcing), leading to over-correction of the model initial condition. The improvement of prediction skill of surface currents by the multi-data assimilation of all the available observations was also reported by Sperrevik et al. (2015).

The ability of the assimilation of ocean surface currents to correct the position of a SST front in a regional simulation was demonstrated by Yu et al. (2012). In their experiments, they assimilated daily-averaged maps of HF radar derived surface cur-

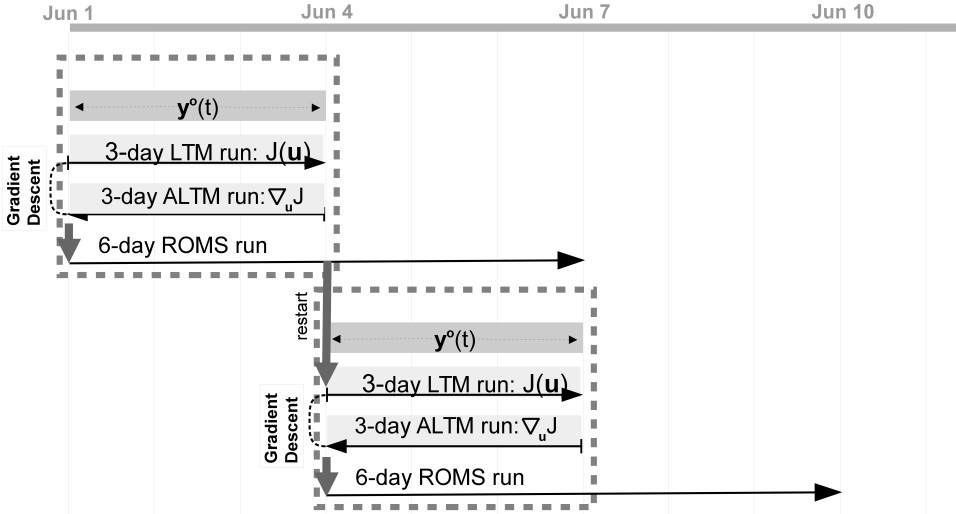

**Figure 14.** Data assimilation cycle in Yu et al. (2012). The data assimilation is done with the help of a linear tangent model (LTM) ans its adjoint code (ALTM). The LTM is an approximation to the linearized dynamics of the ROMS model, used for both the forecast step and to define the reference solution of the LTM model. No overlap between the different assimilation cycles.

rents defined in their 6-km grid. The ocean model was nested inside the 9-km grid Navy Coastal Ocean Model of the California Current System (NCOM). Although ROMS was the ocean model used to simulate the circulation, the data assimilation used a stand-alone linear tangent model (LTM) and its exact adjoint code (ALTM). The LTM was dynamically compatible with the non-linear model and its reference ocean state is obtained by the temporal interpolation of the ROMS trajectory, sampled every
4 hours. With the data assimilation strategy shown in Figure 14, they control the initial condition. After the minimization of the cost function, the initial condition was used to provide a 6-day forecast with ROMS. The model output after three days was used as a first guess for the next assimilation cycle. Although the surface winds were not corrected by the assimilation, it was found that t he assimilation of the HF radar data was able to improve the geometry of the SST front.

Iermano et al. (2016) used the ROMS model and its adjoint to simultaneously assimilate hourly HF radar data in the Gulf
of Naples (Italy), together with 8-day mean product of SST (merging microwave and infrared data) with horizontal resolution of 4.4 km, and daily absolute dynamic topography with horizontal resolution $1/8°$. The simulation domain corresponded to the Tyrrhenian Sea. The control $u$ of the cost function 60 were the initial conditions, the surface forcing and the open boundary conditions. The assimilation window was 7 days. Despite the significant variability between assimilation cycles, the reconstructed circulation was able to correct the location of ocean features as submesoescale jets near the region covered by
the HF radar (figure 15).

Finally, the work of Phillipson and Toumi (2017) assesses the added value of assimilating OSCAR velocity fields in their forecasting system of the Angola Basin circulation. Their baseline experiment assimilates satellite sea surface temperature, and in situ profiles of temperature and salinity. Gridded sea surface height (available daily), OSCAR velocity fields (available every

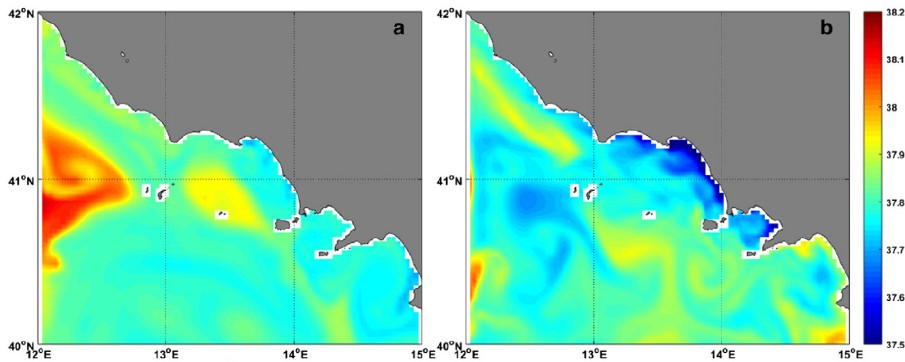

**Figure 15.** Surface salinity field (daily average) corresponding to November 14, 2010 without (left) and after (right) assimilation. From figure 1 in Santoki et al. (2013).

| Technique | Velocities | Latency | $\Delta x_{grid}$ | $\Delta x_{min}$ | Section |
|---|---|---|---|---|---|
| Altimetric maps | geostrophic | ~3 days | 30 km | ~75 km | 2.1 |
| Wind stress* | ageostrophic | < 2 h | 12.5 km | ~75 km | 2.2 |
| Feature tracking | total | < 4 h | 20 km | > 20 km | 2.3 |
| Heat equation* | total | < 4 h | 4-16 km | 4-16 km | 2.3 |
| PV inversion | geostrophic | < 4 h | 1 km | ~ 5 km | 2.4 |
| HF Radar WERA | total | 1 h | 200 m | 10-25 km | 3 |
| HF Radar SeaSonde | total | 1 h | 200 m | 40-60 km | 3 |

**Table 1.** Summary of characteristics for the different methods. The latency of altimetric maps is taken to be 3-days, which corresponds to the intermediate map generated by SSALTO/DUACS system although preliminary data is available within 12h (AVISO Altimetry, 2016). The resolution and latency of wind-driven currents is taken from the characteristics of present scatterometer data.

* If these techniques are combined with altimetric maps, their characteristics are those of altimetry

five days) and drifter velocity observations (derived from 6 hourly interpolated drifter positions) have been subsequently assimilated. Their results indicated that drifter velocity assimilation improved Lagrangian predictability. Assimilation of OSCAR improved Lagrangian predictability as much as altimetry but only by half as much as the drifter improvement. However, simultaneous assimilation of drifter and OSCAR velocities degraded the results obtained by assimilating drifter velocities alone.

5   The main reason of the negative impact of OSCAR data was hypothesized to be the low resolution (spatial and temporal) of the velocity field, together with a large spatial coverage, which weighted the assimilation results to such a less accurate estimate of the surface velocity.

## 5   Summary

The retrieval of surface velocities remains one of the most challenging problems in oceanography with an impact in almost all fields of oceanography. At present, the routinely retrieval of ocean velocities at global scale are based on measurements of the Sea Surface Height (SSH) done by altimeters, which are then used to derive surface currents invoking the geostrophic approximation. This is a robust approach, it is an all-weather, global and well understood methodology that has become the standard for oceanographic research and has had a deep impact in our vision and understanding of ocean dynamics. Moreover, the inclusion of information from wind and, more recently, waves, as well as, corrections to the geostrophic approximation provides very realistic estimations of surface ocean currents. Nevertheless, altimetry is limited by the sampling characteristics and noise level of current altimeters implying constrains to observe structures smaller than 75 km or close to the coast. As a consequence, a significant part of the mesoscale field, particularly in those areas with small Rossby radius such as the Mediterranean sea cannot be observed. In addition, operational applications of altimetric maps are limited by the latency of altimetric data and the need of past and future data to generate altimetric maps. Wind-driven currents derived from wind measurements, on the contrary, have very low latency and, potentially, higher spatial resolution. At present, the existence of several scatterometers provides quite good sampling although all points on the Earth surface are not yet covered every 6h. It is worth mentioning that inertial currents are difficult to retrieve due to the lack of information about its phase.

The limitations of altimetric maps has motivated the use of Sea Surface Temperature (SST) observations to obtain surface velocities. Standard methods (feature tracking, inversion of heat equation) require a sequence of SST (or BT) images, which may be difficult to obtain if infrared observations are used. Furthermore, the need of high resolution data for techniques such as the Maximum Cross Correlation (MCC) technique and the low quality of the resulting velocities further limits its operational use. During the recent years the the Surface Quasi-Geostrophic (SQG) framework has emerged as a potential complement to altimetric maps due its high resolution and low latency (see table 1). This approach is able to capture ocean structures of the order of 5-10 km and at distances to the coast of the order of a few km. One of its main limitations, in addition to the presence of clouds, is the need that SST be a proxy of interior Potential Vorticity. Observations and the analysis of numerical models show that this situation is typically found in winter. Nevertheless, velocities derived from SQG could have a strong potential for operational applications, if expert supervision can be done. In addition, its capability to provide surface currents close the coast opens the door to extend the coverage of the currents provided by HF radars and provide a theoretical framework to improve the assimilation schemes.

A large effort is also being devoted to the direct measurement of ocean currents using remote sensing techniques based on the measurements of the Doppler shift. Two complementary approaches are underway: the use of satellite platforms (e.g. SAR) and the use of land-based systems such as HF coastal radars. Presently, the main constraint of these systems is their limited sampling characteristics, which restrict them to case studies. Nevertheless, they do provide insight about the expected contribution than the assimilation of ocean currents will provide to operational oceanography. Although various approaches have been successfully used to use observations of ocean currents to partially constrain non-linear simulations of various coastal areas, and even improve the geometrical location of the temperature fronts, it has been shown that multiple data sources need

to be simultaneously assimilated to better constrain the hydrography of the system. In addition, as the main source of errors in these simulations, advanced multivariate methodologies (ENKF or 4DVAR) need to be used to be able to retrieve wind stress information from ocean currents to further increase the prediction skill of coastal operational systems.

*Acknowledgements.* This work has been funded by the European Space Agency through the GlobCurrent Data User Element project (4000109513/13/I-LG) and the Spanish Ministry of Research through the COSMO (CTM2016-79474-R, MINECO/FEDER, UE) and PROMISES projects. Financial support by Fundación General CSIC (Programa ComFuturo) is also acknowledged. We would like to thank Prof. I. Barton for providing the velocity field obtained through the MCC method. We are appreciative of the comments and advise provided by G. Quartly, B. Chapron and F. Ardhuin and the anonymous reviewers that had helped to improve this review. The authors wold also like to thank the organizing committee of the NLOA for inviting J.I-F and J.B-P, which generated this review.

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
