# Peer review of "Remote sensing of ocean surface currents: A review of what is being observed and what is being assimilated"

_Nonlinear Processes in Geophysics, 2017_

## Short Comment (SC1) · 6 Apr 2017

Dear colleagues, In such a broad review it is difficult to be accurate on each single aspect, and I generally commend the authors for their work. Here are a few ideas about section "2.2 Ageostrophic currents: wind and waves" that the authors may find relevant to incorporate.

1) Writing equation (5) without defining the "total velocity field" is a bit hard. In fact, this form of the equation was first used by Jenkins (Deut. Hydr. Zeit. 1989), and he defined $v_0$ as the quasi-Eulerian velocity, i.e. the Lagrangian mean velocity minus the Stokes drift.

Indeed it is customary to average the momentum equations over the phase of wind-waves that have periods shorter than 30 s, and it is the residual wave motion known

as Stokes drift (Stokes 1847) that appears in the tracer transport equation and some forms of the momentum equations (see Lane et al. JPO 2008, Bennis et al. Ocean Modelling 2011).

2) the role of the Stokes x Coriolis term of eq. (5) has been discussed in the litterature and it may be interesting to note the paper by Rascle and Ardhuin (JGR 2009) in which, contrary to Polton et al. (2005), a realistic time-evolving wave field and stratification was taken into account to interpret the upper ocean currents recorded in the LOTUS3 experiment.

3) Mentioning equation 12 is a disgrace. Monochromatic waves do not exist in the ocean and we know that for random waves the Stokes drift is the sum over the wave spectrum (Kenyon 1969), giving very different surface values, not just profile. In practice a simplified parameterization as a function of wind speed and wave height can be found in appendix C of Ardhuin et al. (JPO 2009), and the surface Stokes drift is generally of the order of 1 to 1.4 times the wind speed.

4) Indeed, as stated on line 20, wave models may be a good source of Stokes drift estimates, but these estimates vary widely with model parameterizations (again see Figure in appendix C of Ardhuin et al. JPO 2009, and also Figure 6 and Table 2 in Rascle and Ardhuin, Ocean Modelling 2013).

5) It could be mentionned about HF radars, that these radar-derived currents do contain most of the Stokes drift (broche et al. 1983, see also Ardhuin et al, JPO 2009). Just like any surface tracer, even SST (Chevalier et al. RSE 2014, http://dx.doi.org/10.1016/j.rse.2013.07.038).

References: Memo. 509, ECMWF, 29 pp. Broche, P., J. C. de Maistre, and P. Forget, 1983: Mesure par radar décamétrique cohérent des courants superficiels engendrés par le vent. Oceanol. Acta, 6, 43–53.

---

## Referee Comment (RC1) · Anonymous Referee #1 · 8 May 2017

General comments

The proposed piece of research makes an overview of the status of the retrieval and assimilation of surface ocean currents, being covered in sections 2 and 3 respectively. Section 2 discusses 5 different approaches to estimate ocean currents from satellite derived products such as SST or SSH. The techniques covered are grounded in various areas such ocean dynamics, non-linear physics or image processing. Section 3 discusses various methods to assimilate measured ocean currents, namely: nudging, ensemble Kalman Filters and 4DVAR.

The document is not suited for publication at its present form and needs major revisions.

Also, the variety of aspects covered in this paper forces it to overview each topic and,

necessary, overlooking some specific details. I do not have any issue with this approach if the editors find it acceptable and that is within the scope of this special number.

Specific comments

1 – The document lacks coherence giving the feeling that is a collection of separated texts and not part of a structured discussion. This is partially reflected in the parts of the text used as introductions, which are vague and do not properly describe the contents that follow. Last paragraph of Section 1 can be extended to give more information about the aspects covered in the paper. Introduction for Section 2 only describes sections 2.1, 2.2 and 2.5. Section 2.4 is mention but nothing is said about the methodology described and 2.3 is omitted. Introduction for section 3 has no relation with any of the following sub-sections as there is no mention to HF radars or assimilation methods.

More importantly, the last phrase of the abstract suggests that the ocean currents obtained with the methods described in section 2 are going to be then the examples for the assimilation methods described in section 3. However, all examples from section 2 refer to large scale current estimations while section 3 describes the assimilation of HF currents, which are confined to areas close to the shore. This aspect gives the paper a feeling of disconnection between section 2 and 3 that needs to be addressed. That can be either clearly describing and justifying this approach in the appropriate sections of the text (abstract, introduction... etc) or providing data assimilation applications with currents obtained with the methods described in Section 2.

2 – I do acknowledge that it is simply impossible to cover all the aspects of the methods described by the paper. However, it would be good to mention which are relevant and are not possible to cover. Here I outline some examples but I encourage the authors to indicate the ones they consider more relevant based on their expertise. For example:

i) The estimation of the error of a satellite derived product is important to have a measure of the confidence on the data. This is particularly important if the data is going

to be used for data assimilation applications, where an accurate specification of the observation error covariance matrix (R) is critical. Authors indicate which sources of information might be more prone to have high errors, but no indication on how estimate them is given.

ii) The background error covariance matrix Pf, estimated by EnKF methods usually sufferers from an under sampling problem (off diagonal terms are noisy due to the fact that not enough ensemble members are used). To overcome this some localisation needs to be applied to this matrix. May be something about this can be mentioned in the text?

iii) The estimation of the B matrix for 4DVAr algorithms is a non-trivial problem. May be some methodologies can be indicated?

3- Some parts of the text have a feeling of urgency, with confusing phrases and typos, while others are well written in a language that is clear and easy to follow. May be more time can be spent in correcting this before sending the document to the next revision interaction?

I have indicated all the typos I have found in the comments section below. For some of these typos is difficult to understand how they were allowed in the presented version of the manuscript.

4 - Section 3.1 (page 20, line 8) feels more like part of the introduction for section 3. Authors may want to consider appending it to the introduction instead of having it as a separate sub-section.

5- I urge the authors to review the description of the "innovation vector" and the "K" matrix at page 23 (lines 2 to 6), as it seems particular non-standard. To my understanding the "innovation vector" represents the departures between the observations and the model converted to the observations space. "K" represents the weighs of the linear combination between model and observation defined by the values of Pf and

R. Finally, the term K[y-Hx] represents the increments that applied to the background field, gives an optimal analysis provided Pf and R.

Technical comments

P1L3 – "synoptically at global scale" -> "globally at synoptic scale" perhaps more appropriate?

P1L18, P14L9, P14L15, P17L24, P19L1 – It seems awkward to use "on the other hand" without a preceding phrase with "on one hand". May be "Conversely" or "On the contrary" can be considered?

P1L22 to L24 – I suggest to re-phrase as: "For example, coastal HF radars are able to resolve rapid changes and, although the number of HF radars has rapidly increased in the last decades, their coverage remains limited".

P1L25 – Short statement about a new topic that is then not mentioned again. Perhaps more can be said about moorings.

P2L5 – "inn" -> "in"

P2L7 – "acoustic currentmeters" have not been introduced. Are the ones at L4? If so, please clarify.

P2L8 – coma missing after "zone"

P2L11 – "Rossby radius of deformation" is, may be, a more common wording?

P2L12 – Inaccurate, latitude increases from South pole to North pole. Please rephrase.

P2L16 – 1996 seems old for a reference about the past efforts in ocean data collection.

P2L20 – "resulting climatological fields" suggests that it is immediate to obtain them from observations. I would rephrase indicating that the climatological fields are calculated with the observations, sometimes using numerical models and data assimilation to provide a physical coherence for the gaps.

P2L20 – Phrase starting with "These..." seems to be out of place within the ongoing discussion. Consider rephrase.

P2L24 – "Their" -> "His"

P2L27 – "current" -> "currents"

P2L32 – "such" missing after "approaches"

P3L5 – Two "gain" in the same phrase.

P3L8 – "Remote sensing techniques" -> "satellite products" seems more appropriate

P4L1 – "Dppler" -> "Doppler"

P4L2 – Phrase starting with "Some of..." is confusing, please rephrase.

P4L3 to L5 – "Measurement" is used three times in the same phrase, please rephrase.

P4L10 – "(see section 2 and 2.2)", this is actually section 2. Please correct or rephrase (i.e" this section")

P4L13 – The equation is wrong ("L" should be below), please correct . Also, include in the numbering system.

P4L16 – Please, number the equation.

P4L19 – "later" -> "latter", "include" –> "includes"

P4L20 – Phrase starting with "It is worth..." is confusing, please rephrase.

P6L9 – Not sure "evident" is the appropriate word. May be "depicted"?

P6L12 – "two-side effect" suggest that the two sides described oppose each other, which is not the case. Please, consider rephrasing.

P6L22 – Please, indicate what is the "fast evolving structure at the Alboran Sea".

P9L8 – "wind-induces" -> "wind-induced"

P12L14 – It is not clear if multifractal formalism was or was not formulated in more geometrical terms. Consider rephrasing.

P14L2 – "suggest" –> "suggests"

P14L12 – I would use "algorithms" (in plural).

P14L14 – "corrention" -> "correction"

P14L16 – "sionce Ocean" -> "since ocean"

P14L17 – "have" -> "having"

P15L14 – "the" is duplicated

P15L21 – Phrase starting with "This could..." sounds speculative, consider rephrasing. Perhaps it can be started with "We speculate that..." if that is what authors mean.

P17L5 – "we assume", perhaps "it is assumed" is more appropriated?

P18L17 – "for the first time" is redundant, please remove. It is understood that cited papers indicate novel research.

P19L22 – I would remove the first "small".

P19L23 – Is it, may be, "km" -> "m"?

P19L25 – "defotmation" -> "deformation"

P21L31 – "thei" -> "the"

P21L33 – "to" is duplicated

P22L22 – Not all the terms of the eq are described in the following paragraph. Particularly, matrices R and Pf. Please correct.

P22L25 – "vecor" -> "vector". Actually, all the following occurrences are wrong (more than 10) which made me consult three dictionaries to ensure that "vecor" wasn't an

accepted variant of "vector". Please, correct.

P22L28 – The comment about the notation seems pointless from the mathematical point of view. In any case, a different letter "y" is used to highlight the fact that "x" indicates a vector in the model space and "y" indicates a vector in the observation space.

P23L9 – "covariance matrix" –> "error covariance matrix".

P23L12 – "covariance matrix" –> "error covariance matrices".

P23L20 – Alpha is also known as the "inflation factor" and is needed because EnKF methods tend to be underdispersive and lose spread cycle after cycle. There for, an "inflation factor" is needed to make up for the loose of spread. Consider rephrasing.

P24L10 – "Vessel Traffic Service" case has not been introduced. Does it come from Breivik and Saetra (2001)? If so, please indicate it in the text.

P26L26 – "control variance B" -> "model error covariance B". Also, this matrix has the same meaning as Pf in the EnKF. Please, indicate it in the text.

P27L11 – "o f" -> "of"

P28L12 – "t he" -> "the"

P29L9 – "S AR" -> "SAR"

P29L20 – "hydrograpy" -> "hydrography"

P29L20 – "i n" -> "in"

---

## Referee Comment (RC2) · Anonymous Referee #2 · 17 May 2017

General coments:

The paper cannot be published in its present form. Major corrections are required.

The paper presents a valuable review of the state of the art of two different topics:

• Retrieval of non-coastal ocean current information derived from satellite data • Assimilation of HF coastal current in operational ocean models

The contents are generally well explained, and demonstrate a very good knowledge of the authors in the topics. Furthermore, given the importance of the problems treated, and the difficulty to obtain this kind of updated information on the state of the art, the idea behind the paper is valuable.

Nevertheless, the paper has some important problems that should be tackled to fully

unleash its potential.

Specific comments:

1) There is a clear lack of connection between the two main sections of the paper. One is dealing with global non-coastal currents derived from altimeter, while the other is dealing with data assimilation, but only from coastal HF radar currents. These two topics could be perfectly in separated papers. It is necessary to provide more coherence to the paper to avoid the feeling of two different papers pasted together. The easiest way would be to review the state of the art of assimilation from global currents into numerical models… but unfortunately, that authors already claimed that there is no successful exercise in this line. Another possible link is to review any possible work comparing altimeter derived data with HF currents, providing a link between these two worlds. If all the previous fails, the authors should reflect this dual nature of the paper both in the title and in the introduction, or split in two the paper.

2) Section 2 is failing to provide a pragmatic and consistent overview of the usefulness and validity of the techniques that are being described. For example, for some techniques the limitations are explained in much more detail than for others. It would be highly valuable to define, in a systematic way, the expectations of each technique, as well as its limitations in terms of accuracy, capability of deliver timeliness information, spatial resolution, etc…

In this sense, and being a review paper, it is obvious than additional information should be included on the pros and cons of these techniques when compared to the other main source of current information, the operational forecast models.

Finally, given the nature of the paper (a review by experts) some insight should be included on the value of the present techniques to address different specific problems, that at the end are linked with different spatial and temporal scales. Maybe some of the techniques are not valid for some uses like, for example, oils spill forecast, but could be very useful to derive a climatology. This is never addressed, and is vital.

A possible solution to most of these problems could consist on a table explaining, for each one of these techniques, the status of development, limitations and possible uses.

3) Inertial currents are in some occasions and during given time windows the main contribution to ocean currents. Nevertheless, seem like the different retrieval methods are not able to deal with this component. If this is the case, additional assessment should be included.

4) The mathematical formulation in section 2 seems to be in some occasion excessive and unjustified by the text (i.e. reference to Rossby number to define what is geostrophic and ageotropic contributions. Another point where this can be observed is in the description of ageostrophic velocities that lead to expression 16. This formula is obtained just to inform the reader some lines further than the connection is done in practice by adjusting with surface drifters.

5) Section 2.3 seems disconnected with the rest of the chapter. It is not retrieving currents, but providing streamlines. I recommend to move it to the end of section 2, including it as a part of section 2.4 (that would be converted in 2.3), and be treated as a bonus derived from analysis of data imagery (not a as a current retrieval method with its own section)

6) Section 3 should improve the information on how much improvement is expected from the different data assimilation methods. For example, it is stated that some methods improve the position of the fronts, but it is no explained properly how much. In this sense, selected figures with results should be include in a paper of this nature, providing both a more pleasant reading experience and a better insight of the benefits derived from data assimilation.

---

## Author Comment (AC1) · 25 Jul 2017

**Responses to Referee #1:**

Specific comments:

1 – The document lacks coherence giving the feeling that is a collection of separated texts and not part of a structured discussion. This is partially reflected in the parts of the text used as introductions, which are vague and do not properly describe the contents that follow. Last paragraph of Section 1 can be extended to give more information about the aspects covered in the paper. Introduction for Section 2 only describes sections 2.1, 2.2 and 2.5. Section 2.4 is mention but nothing is said about the methodology described and 2.3 is omitted. Introduction for section 3 has no relation with any of the following sub-sections as there is no mention to HF radars or assimilation methods.

To provide a more coherent review as asked by both referees, we have modified the structure of the manuscript and profusely modified the introduction and summary sections. Moreover, to better communicate the two aspects of the review we have modified the title of the manuscript: "Remote sensing of ocean surface currents: A review of what is being observed and what is being assimilated". We have renamed the subsections in Section 2 (see below). The previous section 2.3 " Tracer phase: singularity analysis" has been merged with the previous section 2.5 " Potential vorticity inversion: synergy of sensors" now called "2.4 Currents from a single tracer image". We have introduced a new section 3 called "Retrieval from High Frequency Radars" where we include a short description of this technology for remote sensing of the ocean velocity field and their associated temporal and spatial resolution.

More importantly, the last phrase of the abstract suggests that the ocean currents obtained with the methods described in section 2 are going to be then the examples for the assimilation methods described in section 3. However, all examples from section 2 refer to large scale current estimations while section 3 describes the assimilation of HF currents, which are confined to areas close to the shore. This aspect gives the paper a feeling of disconnection between section 2 and 3 that needs to be addressed. That can be either clearly describing and justifying this approach in the appropriate sections of the text (abstract, introduction, etc) or providing data assimilation applications with currents obtained with the methods described in Section 2.

There are no experiments assimilating global velocity fields as the ones derived in Section 2. There are two regional experiments assimilating OSCAR currents with mixed results. Therefore, most of the experiments assimilating ocean currents correspond to coastal systems. We have modified the title and the introduction to clarify this issue. The Summary outlines some potential options to make the bridge between the open ocean estimates of surface currents with the coastal applications.

2 – I do acknowledge that it is simply impossible to cover all the aspects of the methods described by the paper. However, it would be good to mention which are relevant and are not possible to cover. Here I outline some examples but I encourage the authors to indicate the ones they consider more relevant based on their expertise. For example:

i) The estimation of the error of a satellite derived product is important to have a measure of the

confidence on the data. This is particularly important if the data is going to be used for data assimilation applications, where an accurate specification of the observation error covariance matrix (R) is critical. Authors indicate which sources of information might be more prone to have high errors, but no indication on how estimate them is given.

We agree with you that error estimation is a key issue particularly if you are thinking in assimilating these data. This is an extremely difficult question to answer, specially in the case of remote sensing products. For most of the methods described in Section 2, an estimation of the resulting error depends on many factors, which are not always independent. There are instrumental errors (which in the case of remote sensing is not clear at all mainly due to the lack of in situ validation for many radiometers); representativeness errors (that arise when comparing averaged retrievals with point-wise measurements); interpolation errors (which are a function of the geometry of the sampling and the interpolation methods and parameters); and errors in the validity of the dynamical assumptions, which change in space and time. The manuscript already contains information about the error sources with citations of the published work on this matter. However, we have included an additional comment that summarizes the importance of such an issue in the Summary section.

ii) The background error covariance matrix Pf, estimated by EnKF methods usually suffers from an under sampling problem (off diagonal terms are noisy due to the fact that not enough ensemble members are used). To overcome this some localisation needs to be applied to this matrix. May be something about this can be mentioned in the text?

iii) The estimation of the B matrix for 4DVAr algorithms is a non-trivial problem. May be some methodologies can be indicated?

In the reviewed literature these issues have been dealt differently by different authors. In both cases we have included a statement pointing out each one of these issues in particular.

3- Some parts of the text have a feeling of urgency, with confusing phrases and typos, while others are well written in a language that is clear and easy to follow. May be more time can be spent in correcting this before sending the document to the next revision interaction?

I have indicated all the typos I have found in the comments section below. For some of these typos is difficult to understand how they were allowed in the presented version of the manuscript.

We apologize. The new version of the paper has been inexhaustibly checked. We have tried to correct all the typos.

4 - Section 3.1 (page 20, line 8) feels more like part of the introduction for section 3. Authors may want to consider appending it to the introduction instead of having it as a separate sub-section.

You are right. We have moved part of this section to the introduction and we have rewritten it as a new Section focusing on HF radars.

5- I urge the authors to review the description of the "innovation vector" and the "K"matrix at page 23 (lines 2 to 6), as it seems particular non-standard. To my understanding the "innovation vector" represents the departures between the observations and the model converted to the observations space. "K" represents the weighs of the linear combination between model and observation defined by the values of Pf and R. Finally, the term K[y-Hx] represents the increments that applied to the background field, gives an optimal analysis provided Pf and R.

This part of the text has been completely rewritten in the new version of the manuscript.

Technical comments:

We have completely rewritten the text and most of the following comments are no longer valid although we took all of them into consideration. In what follows you will find those comments that are still relevant for the content of the version.

P1L3 – "synoptically at global scale" -> "globally at synoptic scale" perhaps more appropriate?

After consideration of your suggestion we have modified the statement as follows: "First, no observing system is able to provide direct measurements of global ocean currents at synoptic scales."

P1L18, P14L9, P14L15, P17L24, P19L1 – It seems awkward to use "on the other hand" without a preceding phrase with "on one hand". May be "Conversely" or "On the contrary" can be considered?

The mentioned uses of "On the other hand/side" have been modified as follows: P1L18: "Furthermore"; P14L9: "However, while ..."; P14L15: "With respect to the chlorophyll concentration"; P17L24: (removed); P19L1: "Conversely".

P1L22 to L24 – I suggest to re-phrase as: "For example, coastal HF radars are able to resolve rapid changes and, although the number of HF radars has rapidly increased in the last decades, their coverage remains limited".

Thanks. We have modified the statements according to your suggestion.

P1L25 – Short statement about a new topic that is then not mentioned again. Perhaps more can be said about moorings. P2L7 – "acoustic currentmeters" have not been introduced. Are the ones at L4? If so, please clarify.

In the introduction, for completeness, we have made a historical overview of the technologies used to measure ocean currents and mooring-based instruments mentioned as a key source of in situ

information, mainly in the past. Nevertheless, the focus of the paper is on remote sensing retrieval of surface currents surface currents where moorings play a relative minor role specially with respect the spatial resolution. We have added a new figure (figure 3) comparing the capabilities of each observational technology to measure sea surface currents (according to the GOOS panel) to highlight the advantages of remote sensing (satellites and HF radars) in terms of spatial and time coverage.

P2L20 – "resulting climatological fields" suggests that it is immediate to obtain them from observations. I would rephrase indicating that the climatological fields are calculated with the observations, sometimes using numerical models and data assimilation to provide a physical coherence for the gaps.

To better focus on the goal of the review we no longer talk about "climatologies"

P4L13 – The equation is wrong ("L" should be below), please correct . Also, include in the numbering system.

We have corrected the equation and now corresponds to equation number 1.

P6L22 – Please, indicate what is the "fast evolving structure at the Alboran Sea".

We have modified the statement in the new version.

P19L23 – Is it, may be, "km" -> "m"?

We refer to hundreds of kilometers. It has been written explicitly to avoid confusion.

P22L22 – Not all the terms of the eq are described in the following paragraph. Particularly, matrices R and Pf. Please correct.

The missing descriptions have been added.

P22L25 – "vecor" -> "vector". Actually, all the following occurrences are wrong (more than 10) which made me consult three dictionaries to ensure that "vecor" wasn0t an accepted variant of "vector". Please, correct.

We apologize. All this has been corrected.

P22L28 – The comment about the notation seems pointless from the mathematical point of view. In

any case, a different letter "y" is used to highlight the fact that "x" indicates a vector in the model space and "y" indicates a vector in the observation space.

Rephrased and the text has been shortened.

P23L9 – "covariance matrix" –> "error covariance matrix".

P23L12 – "covariance matrix" –> "error covariance matrices".

Added.

P23L20 – Alpha is also known as the "inflation factor" and is needed because EnKF methods tend to be underdispersive and lose spread cycle after cycle. There for, an "inflation factor" is needed to make up for the loose of spread. Consider rephrasing.

Rephrased: "The parameter $\alpha$, known as \textit{inflation factor}, is introduced to scale the weight of the ensemble versus the observations, to take into account the effect of the model error, and to avoid the collapse of the covariance matrix."

P24L10 – "Vessel Traffic Service" case has not been introduced. Does it come from Breivik and Saetra (2001)? If so, please indicate it in the text.

Rephrased: "The low cost of the EnOI made possible to have a 6-hour forecast within 45 minutes since the data acquisition time."

P26L26 – "control variance B" -> "model error covariance B". Also, this matrix has the same meaning as Pf in the EnKF. Please, indicate it in the text.

We do not agree. The control variance is the same as the model error covariance only when the control vector is the initial condition. If the control vector contains variables or parameters other than the initial condition, the control variance differs from the model error variance. To avoid confusion, we have added the following text: "Note that if the initial model state is the only control variable, then control variance matrix $\vec{B}$ should be equal to the model error covariance $\vec{P}^f$ used in the EnKF."

---

## Author Comment (AC3) · 25 Jul 2017

**Response to the comments done by Dr. F. Ardhuin**

Dear colleagues, In such a broad review it is difficult to be accurate on each single aspect, and I generally commend the authors for their work. Here are a few ideas about section "2.2 Ageostrophic currents: wind and waves" that the authors may find relevant to incorporate.

1) Writing equation (5) without defining the "total velocity field" is a bit hard. In fact, this form of the equation was first used by Jenkins (Deut. Hydr. Zeit. 1989), and he defined v0 as the quasi-Eulerian velocity, i.e. the Lagrangian mean velocity minus the Stokes drift.
Indeed it is customary to average the momentum equations over the phase of wind- waves that have periods shorter than 30 s, and it is the residual wave motion known  as Stokes drift (Stokes 1847) that appears in the tracer transport equation and some forms of the momentum equations (see Lane et al. JPO 2008, Bennis et al. Ocean Modelling 2011).

We have modified the text and we have clarified this point in the new version

2) the role of the Stokes x Coriolis term of eq. (5) has been discussed in the litterature and it may be interesting to note the paper by Rascle and Ardhuin (JGR 2009) in which, contrary to Polton et al. (2005), a realistic time-evolving wave field and stratification was taken into account to interprent the upper ocean currents recorded in the LOTUS3 experiment.

We have rewritten this point and we have included a reference to paper by Ardhuin et al (JGR 2009).

3) Mentioning equation 12 is a disgrace. Monochromatic waves do not exist in the ocean and we know that for random waves the Stokes drift is the sum over the wave spectrum (Kenyon 1969), giving very different surface values, not just profile. In practice a simplified parameterization as a function of wind speed and wave height can be found in appendix C of Ardhuin et al. (JPO 2009), and the surface Stokes drift is generally of the order of 1 to 1.4 times the wind speed.

We agree with Dr. Ardhuin that a monochromatic wave is an idealization. Nevertheless, due its simplicity and its use for some applications we have decided to maintain it. However, we have followed the suggestions of Dr. Ardhuin and we have included the proposed reference and we have underlined the importance of taking into account the full spectrum of waves.

4) Indeed, as stated on line 20, wave models may be a good source of Stokes drift estimates, but these estimates vary widely with model parameterizations (again see Figure in appendix C of Ardhuin et al. JPO 2009, and also Figure 6 and Table 2 in Rascle and Ardhuin, Ocean Modelling 2013).

We have included this point in the new version of the paper as well as the proposed references.

5) It could be mentionned about HF radars, that these radar-derived currents do contain most of the Stokes drift (broche et al. 1983, see also Ardhuin et al, JPO 2009). Just like any surface tracer, even SST (Chevalier et al. RSE 2014, http://dx.doi.org/10.1016/j.rse.2013.07.038).
References: Memo. 509, ECMWF, 29 pp. Broche, P., J. C. de Maistre, and P. Forget, 1983: Mesure par radar décamétrique cohérent des courants superficiels engendrés par le vent. Oceanol. Acta, 6, 43–53.

This point has been included in the new version of the paper. Based on the existing literature we have seen that it is still an open debate. For example, it has been suggested that HF radar currents include the entire wave-induced Stokes drift (Graber et al., 1997), part of it (Ardhuin et al., 2009) or none of it (Röhrs and Christensen, 2015).

---

## Author Comment (AC2)

**Responses to Referee #2:**

General comments:

The paper cannot be published in its present form. Major corrections are required.

The paper presents a valuable review of the state of the art of two different topics: A) Retrieval of non-coastal ocean current information derived from satellite data; B) Assimilation of HF coastal current in operational ocean models

The contents are generally well explained, and demonstrate a very good knowledge of the authors in the topics. Furthermore, given the importance of the problems treated, and the difficulty to obtain this kind of updated information on the state of the art, the idea behind the paper is valuable.

Nevertheless, the paper has some important problems that should be tackled to fully unleash its potential.

Specific comments:

1) There is a clear lack of connection between the two main sections of the paper. One is dealing with global non-coastal currents derived from altimeter, while the other is dealing with data assimilation, but only from coastal HF radar currents. These two topics could be perfectly in separated papers. It is necessary to provide more coherence to the paper to avoid the feeling of two different papers pasted together. The easiest way would be to review the state of the art of assimilation from global currents into numerical models. . . but unfortunately, that authors already claimed that there is no successful exercise in this line. Another possible link is to review any possible work comparing altimeter derived data with HF currents, providing a link between these two worlds. If all the previous fails, the authors should reflect this dual nature of the paper both in the title and in the introduction, or split in two the paper.

The aim of this manuscript has always been to focus on reviewing two aspects of remote sensing of ocean surface currents. On the one hand, we are reviewing the different approaches that can be used to produce estimates of sea surface currents from remote sensing data (Sections 2 and 3). On the other hand, to review the advances in assimilation of sea surface currents, specifically centered on HF radar in coastal regions which is, up to now, the only source of direct remote sensing current measurements (Section 4). Is is expected that gained experience and the lessons learned from assimilating currents from HF radars can be translated, and applied, to global data assimilation systems if real-time, quasi-synoptic maps of ocean currents were available either from incoming satellite missions or derived from the methods reviewed in section 2. To avoid the false expectations from potential readers we have changed the title of the manuscript and we have rewritten completely the Introduction section to better reflect the dual nature of the review.

2) Section 2 is failing to provide a pragmatic and consistent overview of the usefulness and validity of the techniques that are being described. For example, for some techniques the limitations are explained in much more detail than for others. It would be highly valuable to define, in a systematic way, the expectations of each technique, as well as its limitations in terms of accuracy, capability of deliver timeliness information, spatial resolution, etc. . .

In this sense, and being a review paper, it is obvious than additional information should be included on the pros and cons of these techniques when compared to the other main source of current information, the operational forecast models.

Finally, given the nature of the paper (a review by experts) some insight should be included on the value of the present techniques to address different specific problems, that at the end are linked with different spatial and temporal scales. Maybe some of the techniques are not valid for some uses like, for example, oils spill forecast, but could be very useful to derive a climatology. This is never addressed, and is vital. A possible solution to most of these problems could consist on a table explaining, for each one of these techniques, the status of development, limitations and possible uses.

In the new version we have been careful to provide a balanced account of details for each of the techniques reviewed. Note however that these products are not yet been used in global operational forecasting models.

We have followed your suggestions and we have now added some new material in the sense you mention. Now, a new figure illustrates (figure 3)  the current status in terms of spatial and temporal scales of sea surface currents observations according to the GOOS panel. We have also included in the summary section a table listing some key parameters for future use in operational assimilation systems (latency, resolution,...)

3) Inertial currents are in some occasions and during given time windows the main contribution to ocean currents. Nevertheless, seem like the different retrieval methods are not able to deal with this component. If this is the case, additional assessment should be included.

Inertial currents are the ocean response to the range of atmosphere-ocean interaction processes excited when winds are intermittent. Most of the remote sensing satellite systems are not able to satisfy this requirement because the time resolution needed is not high enough to capture this variability. In fact, that is the main reason why equations 7 and 11, which are the base for many retrieval approaches of sea surface currents, lack the temporal term looking only for steady solutions.

Note however that HF radars are the only systems that attain such high temporal sampling and, in fact, they observe and can resolve both tidal flows (semidiurnal and diurnal) and inertial currents which are within the same range of time scales. In the paper it is mentioned the resolution of the data assimilation of such systems. There are systems that average current data daily, over the inertial period and even assimilate data every 20 minutes. However we did not found specific literature centered on resolving inertial variability.

4) The mathematical formulation in section 2 seems to be in some occasion excessive and unjustified by the text (i.e. reference to Rossby number to define what is geostrophic and ageotropic contributions. Another point where this can be observed is in the description of ageostrophic velocities that lead to expression 16. This formula is obtained just to inform the reader some lines further than the connection is done in practice by adjusting with surface drifters.

In the new version we have simplified the mathematical notation and rewritten section 2: reference to Rossby number has been simplified and clarified but, for the wind and waves section, we have rewritten the text while keeping the logical structure. The reason is that surface currents are very complex and recent advances in trying to infer sea surface currents are now including more and more processes. The situation is similar to the evolution of ocean numerical models that only lately start to implement waves effects, Langmuir circulations and so on in new versions of numerical codes. In our case we opted to first describe classical solutions and then look at the algorithms and procedures able to exploit present observational systems to unveil the complexity of these processes.

5) Section 2.3 seems disconnected with the rest of the chapter. It is not retrieving currents, but providing streamlines. I recommend to move it to the end of section 2, including it as a part of section 2.4 (that would be converted in 2.3), and be treated as a bonus derived from analysis of data imagery (not a as a current retrieval method with its own section)

We have followed your suggestion and made changes accordingly.

6) Section 3 should improve the information on how much improvement is expected from the different data assimilation methods. For example, it is stated that some methods improve the position of the fronts, but it is no explained properly how much. In this sense, selected figures with results should be include in a paper of this nature, providing both a more pleasant reading experience and a better insight of the benefits derived from data assimilation.

In the new version we have included three new figures illustrating the impact of assimilating ocean current data in coastal applications.

---

## Referee Report (RR1)

This version of the manuscript has greatly improved the previous one. The reorganization of the sections and the rewriting of a substantial part of the text has bring to the document a well defined structure and a language that is clear and easy to follow. I have particularly enjoyed the didactic style of some of the parts, which is very appropriate for a review document where the most likely readers will be newcomers to the topic.

I would like to thank the authors for the effort spent correcting what was suggested by the reviewers, as well as other things.

The paper is now close to be accepted for publication, only some minor issues and inconsistencies should be corrected.

I understand that this version of the manuscript has been submitted without highlighting the changes because it has been extensively re-written. I encourage the authors to provide a manuscript highlighting the changes for the next version.

**Minor revisions**

There are some issues in the description of the data assimilation methods

**Section 4**

P24L25 "by partially correcting surface wind forcing". This phrase suggests that DA is correcting the forcing data and DA usually only corrects the model state. Can this be clarified a bit further?

**Section 4.2**

It would be worth mentioning that in EnKF applications the error covariance matrix Pf is almost never calculated explicitly using the formula indicated in (55). Instead, the model is first converted to the observation space and the terms PfHt and HPfHt are then calculated as summations. This substantially reduces the computational cost and removes the need for a Ht operator . Please see eqs 6 and 7 in Houtekamer and Zhang (2017), for a clarification on this.

**Section 4.3**

It should be noted that Hoteit (2009) description of 4DVAR differs from the more common formulation used in several operational atmospheric and ocean implementations. The difference lies in the second term of (59) that is commonly defined at the analysis time (to) and is not time dependant. This formulation ensures that the analysis minimizes the error with the background field at t=0 and with the observations across the time window, and is the reason why only the observation term is time dependant. For a reference on this, please check, Mike Fisher ECMWF lecture notes, eq 4, and Mogensen and Alonso-Balmaseda ECMWF technical note 668, eq (1) (see at the bottom for links to these references). These documents contain descriptions of 4DVAR from atmospheric and ocean implementations as they are used in several operational centres. A more theoretical discussion can be found at LeDimet & Talagrand (1986), which is a 4DVAR classical paper and it would be a nice addition to the paper references. Authors might want to reconsider the proposed formulation for 4DVAR.

The matrix B is named inconsistently as "control variance matrix" (P32L15, P32L18), "covariance matrix" (P32L25, P32L29) and "error covariance matrix" (P33L19). I mentioned in my first review that the B matrix can only be interpreted as the "error covariance matrix". Please check the paper

Hoteit et al (2009), cited at the manuscript and used as a guideline for the section, for a clarification on this. In Hoteit (eq 2, equivalent to eq 59 in the manuscript), the R and B matrices are noted as R and Q and described as: "$R(t)$ and $Q(t)$ and are the covariance matrices of observational and first-guess control uncertainties, respectively". In the context of Hoteit et al (2009), "uncertainties" is equivalent to "errors" or "unkowns" in other formulations. Please, review the manuscript to ensure that the B matrix is noted in the appropriate way.

P32L15, "R and B … are a function of time". While at Hoteit et al (2009) these matrices are defined as time dependant, this is not always the case. In many applications (i. e. operational implementations), R and B are commonly formulated as climatic (static) matrices. I suggest to re-phrase as "R and B … might be time dependant". Also, this is consistent with the way R and B are noted in (59), where their time dependency is not explicitly indicated.

**Comments/ Other**

P1L8 fie ld > fields
P1L10 models > models
P1L19 f ew > few
P4L17 "the number of HF has increased..." this phrase is then repeated at P2L21. Authors might consider rewriting.
P4L27 satellite > satellites
P4L31 "Improvements of a better understanding" sounds redundant. May be "A better understanding" or "Progress in the understanding"?
P5L10 I would add a description of section 2 before describing each of the sub-sections. i.e "Section 2 reviews retrievals from satellite observations..."
P5L10 sea level > sea level measurements
P6L24 directions > direction (or is it "perpendicular directions"?)
P7L8 level > the level
P7L11 strung > strong
P8L12 "this example" repeated in P8L13, please re-phrase.
P8L23 "SST of" > "SST at"?
P8L31 "the residuals to respect along track data", not sure if this part of the phrase is correctly worded.
P9L26 "allowed" followed by "allowing". I am being picky here, probably fine...
P10L5 "which is the Lagrangian mean velocities due to waves". Wrong number. "which are" or "which is ... velocity".
P14L27 wave > waves
P15L24 Ocean > ocean
P19L3 remove "does"
P19L19 "an exponential stratifications". Wrong number "exponential stratifications" or "an exponential stratification"
P20L14 "SQG" or "eSQG"? Please review.
P22L13 Something is missing in the phrase starting with "An alternative to... ". May be "An alternative is necessary to..." or "There is an alternative to..."
P22L14 "to" duplicated.
P24L5 Wrongly formatted reference.
P25L15 wrong section reference
P27L22 "Gain", wrong format
P27L24 "... assimilation increment is used..." > "... assimilation increment and is used..."
P27L26 "defined given", choose one
P27L30 describes > describe

P30L10 Figure, wrong format
P33L3 o f > of
P34L10 After minimization > After the minimization
P34L13 t he > the
P34L17 The phrase "The control … conditions" is incomplete. Please review.
P35L7-8 What is "Lagrangian predictability"? Please review and clarify.
P35L11 "together with a large spatial coverage". Why is this bad? Is it because is impacting a large area of the model? Please, clarify.
P35L21-P36L2 Phrase incomplete, please review.
P36L12 this > This
P36L17 open > opens
P36L17 to > the
P37L6 method ologies > methodologies

Caption Fig 5: "paremeters" > "parameters"

**References**

M Fisher. *Assimilation techniques (4): 4dVar.* ECMWF Lecture Notes. (https://www.ecmwf.int/sites/default/files/elibrary/2002/16933-assimilation-techniques-4-4dvar.pdf)

P. L. Houtekamer, F. Zhang. *Review of the Ensemble Kalman Filter for Atmospheric Data Assimilation.* Monthly Weather Review. December 2016.

F.X. LeDimet, O. Talagrand. *Variational algorithms for analysis and assimilation of meteorological observations: theoretical aspects*. Tellus. 1986

K. Mogensen; W. M. Alonso Balmaseda. *The NEMOVAR ocean data assimilation system as implemented in the ECMWF ocean analysis for System 4*. ECMWF Technnical Note 668. (https://www.ecmwf.int/sites/default/files/elibrary/2012/11174-nemovar-ocean-data-assimilation-system-implemented-ecmwf-ocean-analysis-system-4.pdf)

---

## Referee Report (RR2)

Most of the comments on the first review have been fulfilled by the authors. These changes have made a much better and clear paper.

Particularly appropriate is the change of the title of the paper and changes in the introduction.

1) Unfortunately, the modifications suggested for point 2 have not been properly followed:

> *2) Section 2 is failing to provide a pragmatic and consistent overview of the usefulness and validity of the techniques that are being described. For example, for some techniques the limitations are explained in much more detail than for others. It would be highly valuable to define, in a systematic way, the expectations of each technique, as well as its limitations in terms of accuracy, capability of deliver timeliness information, spatial resolution, etc. . .*
>
> *In this sense, and being a review paper, it is obvious than additional information should be included on the pros and cons of these techniques when compared to the other main source of current information, the operational forecast models.*
>
> *Finally, given the nature of the paper (a review by experts) some insight should be included on the value of the present techniques to address different specific problems, that at the end are linked with different spatial and temporal scales. Maybe some of the techniques are not valid for some uses like, for example, oils spill forecast, but could be very useful to derive a climatology. This is never addressed, and is vital. A possible solution to most of these problems could consist on a table explaining, for each one of these techniques, the status of development, limitations and possible uses.*
>
> *Authors reply:*
>
> *In the new version we have been careful to provide a balanced account of details for each of the techniques reviewed. Note however that these products are not yet been used in global operational forecasting models.*
>
> *We have followed your suggestions and we have now added some new material in the sense you mention. Now, a new figure illustrates (figure 3) the current status in terms of spatial and temporal scales of sea surface currents observations according to the GOOS panel. We have also included in the summary section a table listing some key parameters for future use in operational assimilation systems (latency, resolution,)*

The new table is a step forward in this direction, but part of the problem remains. From the reader point of view, it is difficult to conclude whether a technique is accurate or mature, or just experimental. Without this information, the paper is a more a theoretical review than a review of the state of the art of the methodologies. It would be highly valuable to define, in a systematic way, the expectations of each technique, as well as its limitations in terms of accuracy, operationality, etc. Now thanks to the table, horizontal resolution (Delta x grid vs delta x min must be explained) and latency are clear.

The rest of the ideas by the reviewer in this point (the pros and cons of these techniques when compared to the other main source of current information, the operational forecast models; and insight on the value of the present techniques to address different specific problems) were not followed for this second revision

2) Reading of section 2.2 is still difficult. This section is too long, and I recommend at least sub-sectioning, leaving more clear what is historical development and new approaches.

---

## Author Response (AR2)

**Responses to Referee #1:**

Most of the comments on the first review have been fulfilled by the authors. These changes have made a much better and clear paper.

Particularly appropriate is the change of the title of the paper and changes in the introduction. 1) Unfortunately, the modifications suggested for point 2 have not been properly followed:

*2) Section 2 is failing to provide a pragmatic and consistent overview of the usefulness and validity of the techniques that are being described. For example, for some techniques the limitations are explained in much more detail than for others. It would be highly valuable to define, in a systematic way, the expectations of each technique, as well as its limitations in terms of accuracy, capability of deliver timeliness information, spatial resolution, etc. . .*

*In this sense, and being a review paper, it is obvious than additional information should be included on the pros and cons of these techniques when compared to the other main source of current information, the operational forecast models.*

*Finally, given the nature of the paper (a review by experts) some insight should be included on the value of the present techniques to address different specific problems, that at the end are linked with different spatial and temporal scales. Maybe some of the techniques are not valid for some uses like, for example, oils spill forecast, but could be very useful to derive a climatology. This is never addressed, and is vital. A possible solution to most of these problems could consist on a table explaining, for each one of these techniques, the status of development, limitations and possible uses.*

*Authors reply:*

*In the new version we have been careful to provide a balanced account of details for each of the techniques reviewed. Note however that these products are not yet been used in global operational forecasting models.*

*We have followed your suggestions and we have now added some new material in the sense you mention. Now, a new figure illustrates (figure 3) the current status in terms of spatial and temporal scales of sea surface currents observations according to the GOOS panel. We have also included in the summary section a table listing some key parameters*

*for future use in operational assimilation systems (latency, resolution,)*

The new table is a step forward in this direction, but part of the problem remains. From the reader point of view, it is difficult to conclude whether a technique is accurate or mature, or just experimental. Without this information, the paper is a more a theoretical review than a review of the state of the art of the methodologies. It would be highly valuable to define, in a systematic way, the expectations of each technique, as well as its limitations in terms of accuracy, operationality, etc. Now thanks to the table, horizontal resolution (Delta x grid vs delta x min must be explained) and latency are clear.

We agree with the Reviewer on the importance of such analysis but, as far as we know such study have not been carried on. Moreover, to determine the accuracy, operationallity , etc. of each methodology we are preparing such a systematic study and this review is, indeed, its first step. Here we have collected the existing algorithms that will be systematically tested in a second study using high-resolution numerical simulations.

The rest of the ideas by the reviewer in this point (the pros and cons of these techniques when compared to the other main source of current information, the operational forecast models; and insight on the value of the present techniques to address different specific problems) were not followed for this second revision

We attempted to follow this point but in view of the Reviewers comments we were not successful enough. Indeed, the aim of the expansion of section 5 was to response to this question.

2) Reading of section 2.2 is still difficult. This section is too long, and I recommend at least sub- sectioning, leaving more clear what is historical development and new approaches.

We have reorganized section 2.2 and sub-sectioned it.

**Responses to Referee #1:**

P24L25 "by partially correcting surface wind forcing". This phrase suggests that DA is correcting the forcing data and DA usually only corrects the model state. Can this be clarified a bit further?

The statement now reads: "by partially compensating for the errors existing in the wind forcing"

It would be worth mentioning that in EnKF applications the error covariance matrix Pf is almost never calculated explicitly using the formula indicated in (55). Instead, the model is first converted to the observation space and the terms PfHt and HPfHt are then calculated as summations. This substantially reduces the computational cost and removes the need for a Ht operator . Please see eqs 6 and 7 in Houtekamer and Zhang (2017), for a clarification on this.

You are right that this strong advantage of the EnKF was missing. To include this fact, we have slightly modified equation 56 and included an additional equation. The text now reads

"An advantage of the EnKF is that, at each time step, we can easily calculate the projection of the state vector onto the observation space:

H\vec{X}'(t) = [ H\vec{x}_1(t)-\overline{H\vec{x}}, H\vec{x}_2(t)-\overline{H\vec{x}}, \cdots, H\vec{x}_r(t)-\overline{H\vec{x}} ],

that allows the calculation of the terms $H\vec{P}^f H^\top$ and $\vec{P}^f H^\top$ without the need of explicitly estimating the error covariance matrix $\vec{P}^f$ (equation 55) or the operator $H^\top$ (Houtkeamer and zhang 2016). This fact strongly reduces the computational cost of equation 54."

It should be noted that Hoteit (2009) description of 4DVAR differs from the more common formulation used in several operational atmospheric and ocean implementations. The difference lies in the second term of (59) that is commonly defined at the analysis time (to) and is not time dependant. This formulation ensures that the analysis minimizes the error with the background field at t=0 and with the observations across the time window, and is the reason why only the observation term is time dependant. For a reference on this, please check, Mike Fisher ECMWF lecture notes, eq 4, and Mogensen and Alonso-Balmaseda ECMWF technical note 668, eq (1) (see at the bottom for links to these references). These documents contain descriptions of 4DVAR from atmospheric and ocean implementations as they are used in several operational centres. A more theoretical discussion can be found at LeDimet & Talagrand (1986), which is a 4DVAR classical paper and it would be a nice addition to the paper references. Authors might want to reconsider the proposed formulation for 4DVAR.

The matrix B is named inconsistently as "control variance matrix" (P32L15, P32L18), "covariance matrix" (P32L25, P32L29) and "error covariance matrix" (P33L19). I mentioned in my first review that the B matrix can only be interpreted as the "error covariance matrix". Please check the paperHoteit et al (2009), cited at the manuscript and used as a guideline for the section, for a clarification on this. In Hoteit (eq 2, equivalent to eq 59 in the manuscript), the R and B matrices are noted as R and Q and described as: "R(t) and Q(t) and are the covariance matrices of observational and first-guess control uncertainties, respectively". In the context of Hoteit et al (2009), "uncertainties" is equivalent to "errors" or "unkowns" in other formulations. Please, review the manuscript to

ensure that the B matrix is noted in the appropriate way.

P32L15, "R and B ... are a function of time". While at Hoteit et al (2009) these matrices are defined as time dependant, this is not always the case. In many applications (i. e. operational implementations), R and B are commonly formulated as climatic (static) matrices. I suggest to re-phrase as "R and B ... might be time dependant". Also, this is consistent with the way R and B are noted in (59), where their time dependency is not explicitly indicated.

Thank you for your input. We have modified the notation in the cost function to keep a similar notation as in Hoteit et al. (2009). We have also renamed equation Q as error covariance matrix as suggested. We have also introduced a statement about the fact that many applications use stationary covariance matrices.

As such, we have included two new citations in the manuscript:

[revised manuscript text omitted]